# All-in-one, bio-inspired, and low-power crypto engines for near-sensor security based on two-dimensional memtransistors

Akhil Dodda 1, Nicholas Trainor2,6, Joan. M. Redwing2,3,4 & Saptarshi Das 1,2,3,4,5✉

In the emerging era of the internet of things (IoT), ubiquitous sensors continuously collect, consume, store, and communicate a huge volume of information which is becoming increasingly vulnerable to theft and misuse. Modern software cryptosystems require extensive computational infrastructure for implementing ciphering algorithms, making them difficult to be adopted by IoT edge sensors that operate with limited hardware resources and at low energy budgets. Here we propose and experimentally demonstrate an *"all-in-one"* 8 × 8 array of robust, low-power, and bio-inspired crypto engines monolithically integrated with IoT edge sensors based on two-dimensional (2D) memtransistors. Each engine comprises five 2D memtransistors to accomplish sensing and encoding functionalities. The ciphered information is shown to be secure from an eavesdropper with finite resources and access to deep neural networks. Our hardware platform consists of a total of 320 fully integrated monolayer $MoS_2$-based memtransistors and consumes energy in the range of hundreds of picojoules and offers near-sensor security.

[1] Engineering Science and Mechanics, Penn State University, University Park, PA 16802, USA. [2] Materials Science and Engineering, Penn State University, University Park, PA 16802, USA. [3] 2D Crystal Consortium, Penn State University, University Park, PA 16802, USA. [4] Materials Research Institute, Penn State University, University Park, PA 16802, USA. [5] Electrical Engineering and Computer Science, Penn State University, University Park, PA 16802, USA. [6]Present address: 2D Crystal Consortium, Penn State University, University Park, PA 16802, USA. ✉email: sud70@psu.edu

Information security is key to the sustainable growth and development of any modern society that thrives on global connectivity in this new era of the Internet of Things (IoT). Today, information is collected, stored, and communicated continuously by IoT sensors and edge devices that are found ubiquitously in our homes, workplaces, industrial manufacturing plants, transportation, health sectors, agricultural fields, and so on and so forth. However, with the increasing demand for edge devices, there is an escalating threat of tampering or physical intrusion of privacy by untrustworthy parities[1,2]. While the state-of-the-art cryptosystems offer powerful security solutions based on complex ciphering algorithms that can be implemented using hardware accelerators[3–11], IoT edge devices have many restrictions in terms of computational capabilities due to limited hardware and energy resources. Furthermore, low-cost design needs, large-scale deployments, and the heterogeneous nature of IoT sensors limit the direct adoption of traditional security solutions, including the widely used public-key scheme. Due to inadequate security, IoT devices used in smart cars and smart homes have shown tremendous vulnerability in recent times[12]. Acknowledging these limitations, the United States National Institute of Standards and Technology (NIST) has proposed a multi-year effort, namely, the lightweight cryptography (LWC) standardization process to ensure confidentiality, integrity, and authenticity of data using efficient software modules. However, this effort emphasizes software solutions rather than providing hardware-based technological solutions. Therefore, it is timely to develop on-chip cryptographic primitives that require less computational resources and are integrated with IoT edge sensors.

Here, we exploit the optoelectronic sensing and in-memory compute capabilities of two-dimensional (2D) memtransistors based on photosensitive monolayer $MoS_2$ to introduce near sensor and robust security solutions for IoT edge devices with minimal hardware investments and at frugal energy expenditure. Note that, unlike two-terminal memristors, 2D memtransistors are three-terminal devices, with their additional gate terminal allowing both non-volatile and analog programming of the conductance states as well as electrostatic control of the 2D channel. For our demonstration, we have fabricated an $8 \times 8$ crossbar array of fully integrated crypto engines to encode $8 \times 8$-pixel images. Each engine comprises five monolithically integrated $MoS_2$ memtransistors (5T cell) to accomplish sensing and encoding functionalities. Our entire hardware platform utilizes a total of 320 $MoS_2$ memtransistors, making it one of the very few experimental demonstration of medium scale integrated (MSI) circuits based on 2D materials and devices[13]. In addition, our hardware platform is "self-sufficient," offering all-in-one IoT capabilities that include sensing, compute, storage, and security. Finally, our design inspiration for the 2D memtransistor-based crypto engine is derived from the organization of the peripheral and central nervous systems, which employ similar cell types, i.e., groups of neurons with different functionalities, that transduce external sensory information into electrical impulses to communicate among each other through successive encoding and decoding processes in the presence of a wide range of synaptic noises.

Note that our hardware platform is "self-sufficient," offering all-in-one IoT capabilities that include sensing, compute, storage, and security. This is in sharp contrast to the state-of-the-art silicon-based complementary metal oxide semiconductor (CMOS) technology, which is limited in terms of memory and compute integration owing to the traditional von-Neumann computing architecture. Similarly, non von-Neumann platforms such as field programmable gate arrays (FPGAs) and memristive crossbar arrays lack sensing functionalities. In addition, two-terminal memristors are limited in computational capabilities and require CMOS peripherals, which increases area and energy overhead. In contrast, our proposed three-terminal memtransistor technology with the added feature of photosensitivity owing to the use of monolayer 2D material such as $MoS_2$ offers a stand-alone and holistic solution for in-memory computing and sensing, which is critical for achieving integrated and energy/area efficient security solutions. Note that, beyond photodetectors[14], $MoS_2$ based FETs have been used as chemical sensors[15], biological sensors[15], touch sensors[16], and radiation sensors[17]. Therefore, the sensing unit of our $MoS_2$-based crypto engine is not limited to only optical stimuli. See Supplementary Table 1 for benchmarking of our work against earlier works on 2D materials[18–30], memristors[31–37], phase change materials[38], and nano crystals[39] that combine either "sensing and storage" or "sensing and compute" or "security and storage" but not all aspects in a single hardware platform.

## Results

### All-in-one 2D memtransistor-based hardware fabric.

Figure 1a–c, respectively, show optical images of the fully integrated $8 \times 8$ crossbar array of the crypto engines, a representative crypto engine with five $MoS_2$ memtransistors (5T cell) that combine sensing, storage, and encoding functionalities, and an individual $MoS_2$ memtransistor, which is locally back-gated using a stack comprising atomic layer deposition (ALD) grown 50 nm $Al_2O_3$ on sputter deposited 40/30 nm Pt/TiN. All back-gate islands were placed on a commercially purchased $SiO_2/p^{++}$-Si substrate (see Supplementary Figs. 1–3 for enlarged optical images of the entire chip, $8 \times 8$ array of the crypto engines, and individual crypto engines). As we will discuss later, the $Al_2O_3$/Pt/TiN gate islands not only allow non-volatile programming of our $MoS_2$ memtransistors but also enhance the photoresponse of $MoS_2$ memtransistors owing to the phenomenon of gate-tunable persistent photoconductivity when subjected to the right polarity and magnitude of local back-gate biases. This, in turn, empowers our hardware platform to enable in-memory computing and near-sensor security, which are presently lacking for conventional silicon as well as emerging technologies.

Figure 1d, e, respectively, show the circuit schematic for the crossbar architecture and for each crypto engine (see Supplementary Figs. 4, 5 for enlarged circuit schematics). In short, $MoS_2$ memtransistors used as the photo transistor ($T_{PT}$) mimic sensory neurons and transduce optical information into persistent photoconductance ($G_{PT}$), whereas the $MoS_2$ memtransistors used as the white Gaussian noise adder ($T_{WGNA}$) emulate noisy synapses. Note that $T_{WGNA}$ are pre-programmed into random conductance states ($G_{WGNA}$). As such, $T_{WGNA}$ superimpose white Gaussian noise of finite standard deviation ($\sigma_G$) on the signal transduced by $T_{PT}$ and generate noisy voltage, $V_{PSV}$. Finally, the $MoS_2$ memtransistors used as the spiking neurons ($T_{SN}$) encrypt the noisy presynaptic information i.e., $V_{PSV}$ into post-synaptic current spikes ($I_{PSC}$) using reconfigurable encoding threshold ($V_{ET}$). The remaining two $MoS_2$ memtransistors, i.e., photo selector transistor ($T_{PST}$) and encoding selector transistor ($T_{EST}$) operate as the individual selector switches for the photosensing and the encoding operations, respectively.

In Fig. 1e, $T_{PST}$ and $T_{PT}$ are connected in series at the node $N_3$, $T_{PT}$ and $T_{WGNA}$ are connected in series at the node $N_5$, which is also connected to the local back-gate of $T_{SN}$, and, finally, $T_{EST}$ and $T_{SN}$ are connected in series at the node $N_{10}$. The nodes $N_2$, $N_4$, $N_6$, and $N_9$, respectively, serve as the local back-gate terminals of $T_{PST}$, $T_{PT}$, $T_{WGNA}$, and $T_{EST}$, while node $N_1$ serves as the drain terminal of $T_{PST}$, node $N_8$ serves as the drain terminal of $T_{EST}$, and node $N_7$ serves as the common source terminal for $T_{WGNA}$ and $T_{SN}$. As shown in Fig. 1d, nodes $N_2$, $N_4$, $N_6$, and $N_9$

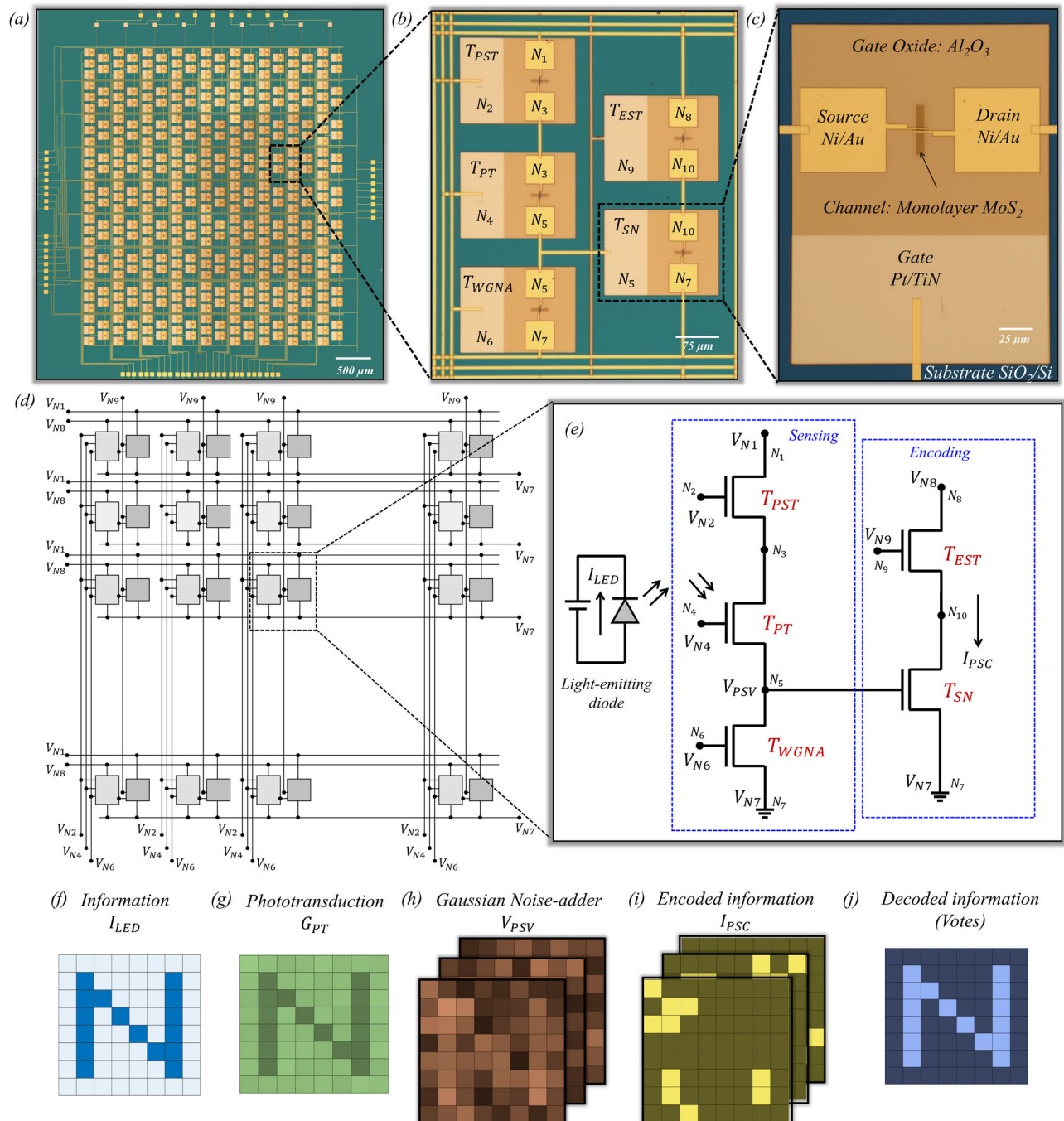

**Fig. 1 *All-in-one* hardware fabric for near sensor security based on multifunctional MoS₂ memtransistors.** Optical images of (**a**) the fully integrated 8 × 8 crossbar array of crypto engines, (**b**) a representative crypto engine with five MoS₂ memtransistors (5T cell) that integrate sensing, storage, and encoding functionalities, and (**c**) an individual MoS₂ memtransistor, which is locally back-gated using a stack comprising 50 nm Al₂O₃ on 40/30 nm Pt/TiN. All back-gate islands were placed on an SiO₂/p⁺⁺-Si substrate. Equivalent circuit diagrams for (**d**) the 8 × 8 crossbar array with column and row select lines and (**e**) each crypto engine. Herein, the MoS₂ memtransistor used as T$_{PT}$ mimics sensory neurons and transduces optical information into persistent photoconductance (G$_{PT}$), the MoS₂ memtransistor used as T$_{WGNA}$ emulates noisy synapses, the MoS₂ memtransistor used as T$_{SN}$ imitates spiking neurons converting noisy presynaptic information, i.e., V$_{PSV}$, into post-synaptic current spikes (I$_{PSC}$), and the remaining 2 MoS₂ memtransistors, T$_{PST}$ and T$_{EST}$, operate as the individual selector switches for the photosensing and the encoding operations, respectively. An example experimental demonstration of sensing and ciphering, where (**f**) the information, for example, an 8 × 8 pixelated image of the letter 'N' obtained by illuminating a blue light emitting diode (I$_{LED}$) is presented to the corresponding 8 × 8 array of the crypto engines. Colormap of (**g**) post-illumination G$_{PT}$, (**h**) noisy V$_{PSV}$, and (**i**) encrypted I$_{PSC}$. **j** The information is revealed by a decoder through a voting process.

from the crypto engines in a given column are connected to common $V_{N2}$, $V_{N4}$, $V_{N6}$ and $V_{N9}$ lines, respectively, and nodes $N_1$, $N_7$, and $N_8$ from the crypto engines in a given row are connected to common $V_{N1}$, $V_{N7}$, and $V_{N8}$ lines, respectively. This allows us to select any crypto engine corresponding to a given row and column. Note that the overlapping connections between the horizontal and vertical metal lines are separated lithographically by depositing an insulating layer of alumina ($Al_2O_3$) at the cross points as described in the Methods section.

Figure 1f–j shows an example experimental demonstration of sensing and ciphering. Note that the proposed security scheme aims at the "confidentiality" of the information. Information, for example, an $8 \times 8$ pixelated image of the letter 'N' (Fig. 1f) obtained by illuminating a blue light emitting diode (LED), is presented one-by-one to the corresponding $8 \times 8$ crossbar array of the crypto engines using the row and column select lines. Figure 1g shows the post-illumination photoconductance map obtained from the corresponding $T_{PT}$, which is transformed into a noisy $V_{PSV}$ map using the respective pre-programmed $T_{WGNA}$, as shown in Fig. 1h, and presented to the corresponding $T_{SN}$ to generate an encrypted $I_{PSC}$ map as shown in Fig. 1i. The information is revealed by a decoder (Fig. 1j) through a voting process when a finite number ($P$) of $8 \times 8$ crypto engines communicate the encrypted information. As we will elucidate in the following sections, the decoding process requires an optimum number of voting mandate ($M_V$) that is determined by $P$, $V_{ET}$, and $\sigma_G$ without the knowledge of which an eavesdropper requires a significant number of brute force trials (BFTs) for deciphering the information. In fact, the information remains concealed even if the eavesdropper has access to a trained deep neural network (DNN). The proposed attack model refers to wiretapping[40].

Note that the monolayer $MoS_2$ used in this study was grown epitaxially on a sapphire substrate using a metal-organic chemical vapor deposition (MOCVD) technique at $1000\,^0C$ by the 2D Crystal Consortium (2DCC). The monolayer film was then transferred using a PMMA-assisted wet transfer process[41,42] from the growth substrate to the target application substrate, i.e., $SiO_2$/$p^{++}$-Si with predefined islands of $Al_2O_3$/Pt/TiN, for subsequent fabrication of the $8 \times 8$ crypto engines. Details on the fabrication of the back-gate islands, monolayer $MoS_2$ synthesis, film transfer, fabrication of $MoS_2$ memtransistors, and monolithic integration can be found in the Methods section. Each $MoS_2$ memtransistor has a footprint ($W \times L$) of $5\,\mu m \times 1\,\mu m$. Given that we have used monolayer $MoS_2$ as the channel material for the memtransistors, which are aggressively scalable, it is possible to reduce the hardware footprint even further.

While any ultra-thin-body semiconductor which is multifunctional, i.e., photosensitive and can be used to fabricate programmable field effect transistors (FETs) or memtransistors, is equally suitable for this demonstration, the use of $MoS_2$ in our *all-in-one* IoT platform has several merits. First, $MoS_2$ is the most advanced among other 2D materials in terms of scalable growth over a large area (wafer scale) as well as in terms of demonstration of high-performance FETs with on current > $250\,\mu A/\mu m$ at ultra-short channel lengths[43] with low device-to-device variability[43–45], which are promising to meet the requirements set forth by the International Roadmap of Devices and Systems (IRDS)[46]. In addition, circuit and architecture level demonstrations of digital, analog, and radio frequency (RF) electronics based on 2D transistors are already available[47–51], and emerging applications such as neuromorphic, optoelectronic, straintronic, hardware security, and biomimetic technologies exploiting the sensing, computing, and storage capabilities of 2D transistors have also been reported[13,26,28,30,52–61]. This work further advances the field of 2D material-based devices by demonstrating robust security features achievable under the same hardware infrastructure used for sensing, storage, and computing. Note that some of our earlier work has also shown the potential use of 2D materials in resolving rampant security vulnerabilities[57,58,61,62].

## Characterization of monolayer $MoS_2$-based memtransistor.

Figure 2a, b, respectively, show the transfer characteristics, i.e., source to drain current ($I_{DS}$) as a function of the local back-gate voltage ($V_{BG}$) at different drain biases ($V_{DS}$), and output characteristics, i.e., $I_{DS}$ *versus* $V_{DS}$ for different $V_{BG}$, of a representative $MoS_2$ memtransistor. Figure 2c shows the device-to-device variation in the transfer characteristics of as-fabricated 64 $MoS_2$ memtransistors used as $T_{SN}$ corresponding to each of the $8 \times 8$ array of the crypto engines (see Supplementary Fig. 6a for the transfer characteristics of each of these 64 $MoS_2$ memtransistors). Figure 2d shows the colormap of electron field effect mobility values ($\mu_{FE}$) extracted from the peak transconductance for these 64 $MoS_2$ memtransistors with a mean of $\sim 8\,cm^2V^{-1}s^{-1}$ and a standard deviation of $3.1\,cm^2V^{-1}s^{-1}$. Supplementary Fig. 6b–d, respectively, show similar colormaps on device-to-device variation in current on/off ratio ($r_{ON/OFF}$), threshold voltage ($V_{TH}$) extracted at an iso-current of $100\,nA/\mu m$, and subthreshold slope ($SS$) over 3 orders of magnitude change in $I_{DS}$, with mean values of $\sim 1 \times 10^6$, $\sim 1.65\,V$, and $\sim 355\,mV/decade$, respectively, and standard deviation values of $\sim 1 \times 10^6$, $\sim 0.42\,V$, and $85\,mV/decade$, respectively. Our $\mu_{FE}$, $r_{ON/OFF}$, $SS$, and $V_{TH}$ values and their corresponding device-to-device variations are on par with the state-of-the-art literature on large area grown $MoS_2$. Also note that, in Fig. 2b, the on-state current in the representative $MoS_2$ memtransistor reaches as high as $\sim 30\,\mu A/\mu m$ at $V_{DS} = 5\,V$ for an inversion charge carrier density of $\sim 1 \times 10^{13}/cm^2$. While on-state performance has minimal impact on the operation of the crypto engine, these metrics confirm high-quality monolayer film growth using MOCVD, relatively damage-free film transfer, and clean device fabrication processes.

Next, we demonstrate the programming capability of our monolayer $MoS_2$ memtransistors in any desirable conductance state with non-volatile memory retention characteristics. Figure 2e, f, respectively, show the post-programmed and post-erased transfer characteristics of a representative $MoS_2$ memtransistor when subjected to negative "Write" ($V_P$) and positive "Erase" ($V_E$) voltage pulses of different amplitudes applied to the local back-gate electrode, each for a duration of $\tau_{P/E} = 1\,s$. The shift in the transfer characteristics can be attributed to charge trapping/detrapping at and near the $MoS_2$/$Al_2O_3$ interface. A negative shift in the transfer characteristics with an increasing magnitude of $V_P$ and positive shift with increasing magnitude of $V_E$ are indicative of electron trapping and de-trapping in the local back-gate stack, respectively. Interestingly, the trapping and de-trapping processes were found to be non-volatile as evident from the retention measurements displayed in Fig. 2g, h for 5 representative post-programmed ($G_P$) and post-erased ($G_E$) conductance states, respectively, for 100 seconds. We also examined long-term memory retention for two representative analog conductance states for $\sim 10^4$ seconds, as shown in Supplementary Fig. 7. The memory ratio ($MR$) between these two states was found to change from $\sim 6 \times 10^2$ to $\sim 2 \times 10^2$ following an exponential decay with a time constant of $7.6 \times 10^3\,s$. The projected time before the $MR$ reaches 1, i.e., these two states become indistinguishable, was found to be $\sim 14\,h$. The $MR$ can be reinstated by reprogramming the devices every few hours or as necessary. This will necessitate some peripheral timing circuits. Improving memory retention through the optimization of our back-gate stack can eliminate the need for

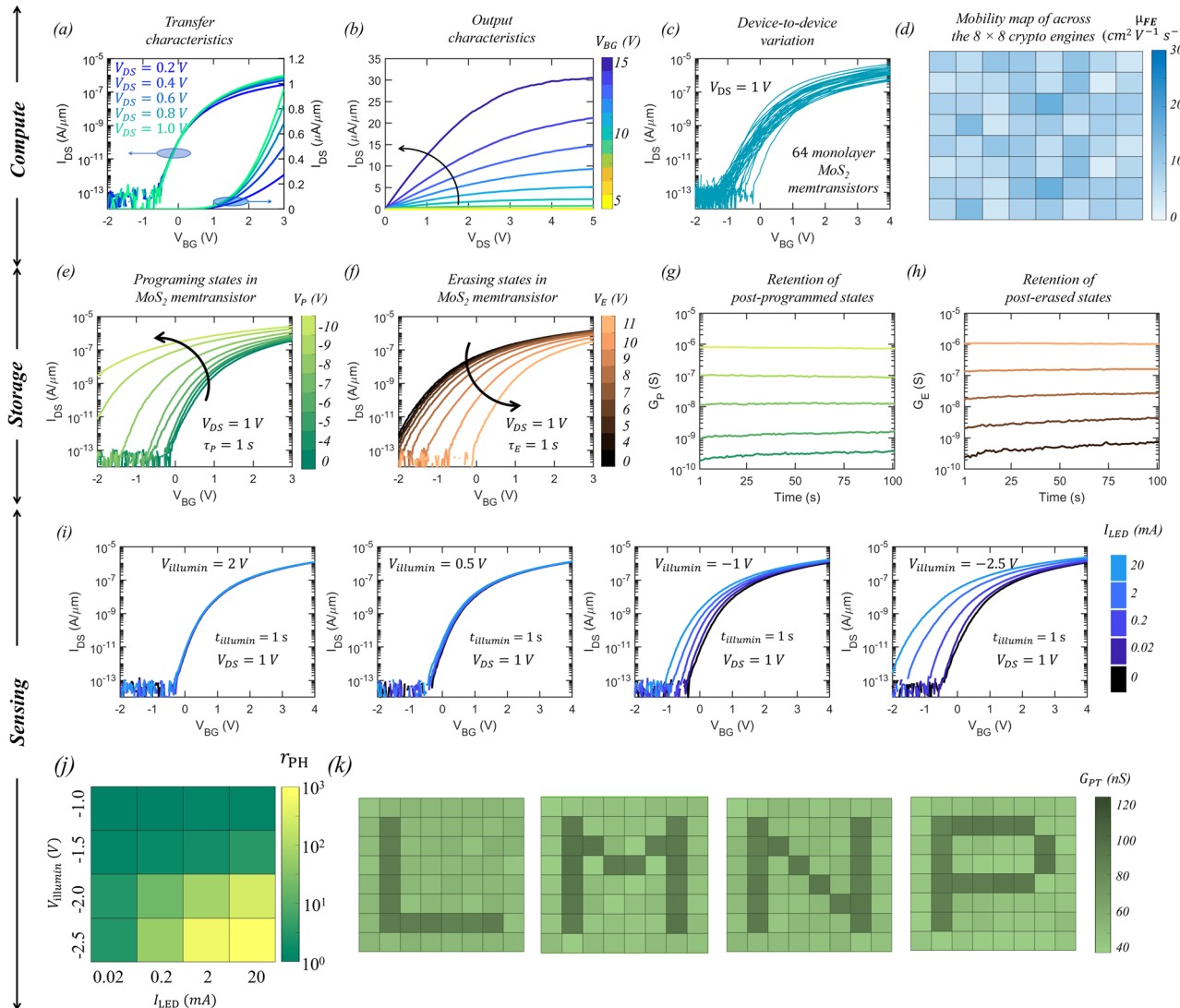

**Fig. 2 MoS$_2$ memtransistor for compute, storage, and sensing. a** Transfer characteristics, i.e., source to drain current ($I_{DS}$) versus back-gate voltage ($V_{BG}$) at different drain biases ($V_{DS}$), and (**b**) output characteristics, i.e., $I_{DS}$ versus $V_{DS}$ for different $V_{BG}$, of a representative MoS$_2$ memtransistor. **c** Transfer characteristics and (**d**) colormap of the distribution of electron field effect mobility values ($\mu_{FE}$) extracted from the peak transconductance of 64 as-fabricated MoS$_2$ memtransistors used as spiking neurons ($T_{SN}$) corresponding to each of the 8 × 8 array of the crypto engines. **e** Post-programmed and (**f**) post-erased transfer characteristics of a representative MoS$_2$ memtransistor when subjected to negative "Write" ($V_P$) and positive "Erase" ($V_E$) voltage pulses of different amplitudes applied to the local back-gate electrode, respectively, each for a duration of 100 ms. Non-volatile retention of 5 representative (**g**) post-programmed ($G_P$) and (**h**) post-erased ($G_E$) conductance states for 100 s. **i** Transfer characteristics for a representative monolayer MoS$_2$ memtransistor before and after illumination from a blue LED with input currents ranging from $I_{LED} = 0.02$ mA (low-brightness) to $I_{LED} = 20$ mA (high-brightness) at different $V_{BG} = V_{illumin}$ for $t_{illumin} = 100$ ms. **j** Colormap of the ratio of post-illumination conductance to dark conductance, $r_{PH}$, extracted at $V_{BG} = 0$ V from (**i**) as a function of $I_{LED}$ and $V_{illumin}$. **k** Colormap of persistent photoconductivity ($G_{PH}$) measured at $V_{BG} = 0$ V when the MoS$_2$ memtransistors used as sensory neurons ($T_{PT}$) corresponding to the 8 × 8 array of the crypto engines are exposed to 8 × 8 pixelated images of the letters 'L', 'M', 'N', and 'P', obtained through the LED illumination for $t_{illumin} = 100$ ms with $V_{illumin} = -2$ V. The bright pixels correspond to $I_{LED} = 20$ mA.

extra peripherals. Similarly, Supplementary Fig. 8 shows the memory endurance for $2 \times 10^3$ cycles for two representative analog conductance states. *MR* was found to change from ~7.5 to ~5 following a power-law decay with an exponent of ~ −0.1. The projected memory endurance before the *MR* reaches 1 was found to be $5 \times 10^8$ cycles, which is comparable to the state-of-the-art FLASH memory devices.

Nevertheless, we exploit the analog nature of the conductance states achievable in MoS$_2$ memtransistors for the realization of noisy synapses, i.e., $T_{WGNA}$, with corresponding $G_{WGNA}$ that follow random Gaussian distribution (in logarithmic scale). Supplementary Fig. 9 shows the device-to-device variation in

the pre and post-programmed transfer characteristics and corresponding colormap of *MR* measured at $V_{BG} = 0$ V for 64 monolayer MoS$_2$ memtransistors corresponding to our 8 × 8 array of the crypto engine when programmed with $V_P = -10$ V for $t_P = 1$ s. The mean and standard deviation values for *MR* were found to be $2.8 \times 10^5$ and $8.7 \times 10^5$, respectively. Note that the inherent device-to-device variation in programming is significantly smaller compared to the look-up-table based variation introduced in the MoS$_2$ memtransistors used as $T_{WGNA}$ in the 8 × 8 array of the crypto engines. Finally, Supplementary Fig. 10 shows the programming and erasing energy expenditures ($E_{P/E}$), which were found to be less than picojoules calculated based on

$E_{P/E} = 1/2 C_G V_{P/E}^2$, where $C_G$ is the capacitance for the local back-gate.

Next, we demonstrate the adaptive photosensing capability of monolayer $MoS_2$ memtransistors by exploiting the unique phenomenon of gate-tunable persistent photoconductivity. Figure 2i shows the transfer characteristics of a representative monolayer $MoS_2$ memtransistor before and after illumination from a blue LED with input currents ranging from $I_{LED} = 0.02$ mA (low-brightness) to $I_{LED} = 20$ mA (high-brightness) at different $V_{BG} = V_{illumin}$ for $t_{illumin} = 1$ s. Note that, instead of standard LASER illumination used to assess the photoresponsivity of monolayer $MoS_2$[18], we have used an LED as the source of external optical stimuli since it represents a more realistic lighting ambience where most edge sensors will be deployed. Figure 2j shows the colormap of the ratio of post-illumination conductance to dark conductance, $r_{PH}$, extracted at $V_{BG} = 0$ V from Fig. 2i as a function of $I_{LED}$ and $V_{illumin}$. Two distinctive types of photoresponse are seen in $MoS_2$ memtransistors. For $V_{illumin} > 0$ V, i.e., illuminations in the on-state and in the subthreshold regime, $r_{PH} = 1$, whereas for $V_{illumin} < 0$ V, i.e., illuminations in the off-state, $r_{PH} \gg 1$. This can be explained from the fact that during illumination in the on-state the photocarriers generated in the $MoS_2$ channel are swept across by the applied $V_{DS}$ and hence there is no persistent photocurrent beyond the optical exposure. However, for $V_{illumin} < 0$ V, persistent photoconductivity emerges as photocarriers get trapped at and near the $MoS_2$/dielectric interface, leading to the shift in the post-illumination device characteristics. The detrapping process can take several hours. Higher $I_{LED}$, more negative $V_{illumin}$, and longer $t_{illumin}$ naturally result in more photocarrier trapping and hence larger shifts in the device characteristics, leading to higher values of $r_{PH}$. Supplementary Fig. 11 shows the colormap of $r_{PH}$ as a function of $V_{illumin}$ and $t_{illumin}$ for $I_{LED} = 20$ mA. Interestingly, $r_{PH}$ values obtained for brighter LED illuminations at less negative $V_{illumin}$ and shorter $t_{illumin}$ can be obtained for dimmer LED illuminations at more negative $V_{illumin}$ and for longer $t_{illumin}$. This phenomenon of gate-tunable persistent photoconductivity can be exploited for adaptive sensing, which is a key requirement for edge sensors. Note that post-illumination $MoS_2$ memtransistors can be reset back to their pre-illumination conditions by applying erase programming voltages, $V_E$, to the local back-gate as described in the context of non-volatile memory capability of our *all-in-one* hardware platform.

Figure 2k shows the colormap of persistent photoconductivity ($G_{PT}$) measured at $V_{BG} = 0$ V when the $T_{PT}$ corresponding to the $8 \times 8$ array of the crypto engines are exposed to $8 \times 8$ pixelated images of the letters, 'L', 'M', 'N', and 'P', obtained through LED illumination for $t_{illumin} = 100$ ms with $V_{illumin} = -2$ V. The bright pixels correspond to $I_{LED} = 20$ mA. Clearly, $MoS_2$ memtransistors integrated with the crypto engines can accurately transcribe the optical information into an electrical response. See Supplementary Fig. 12 for the transcription of $8 \times 8$ pixelated images of the letters 'L', 'M', 'N', and 'P' for dimmer LED illuminations ($I_{LED} = 2$ mA) using different $V_{illumin}$. This demonstration highlights the advantages of gate-tunability of persistent photoconductivity in achieving adaptation to the illumination levels similar to the rod and cone neurons found in the visual system of primates. Finally, Supplementary Fig. 13 shows the pre- and post-illumination transfer characteristics of 64 $MoS_2$ memtransistors used as $T_{PT}$ corresponding to each of the $8 \times 8$ crypto engines and the corresponding colormap of the ratio of post-illumination photoconductance to dark conductance ($r_{PH}$) measured at $V_{BG} = 0$ V. The mean and standard deviation values were found to be $1.6 \times 10^4$ and $3.1 \times 10^4$, respectively. The device-to-device variation in photoconductance had minimal effect on the encryption and decryption process. We would like to remind the readers that, here, "information encryption" refers to the confidentiality of the encoded information.

**Encryption using $MoS_2$ memtransistor.** Next, we illustrate the encryption process by utilizing the sensing, storage, and computing capabilities of $MoS_2$ memtransistors demonstrated above and by using the circuit diagram shown in Fig. 1d, e. Supplementary Fig. 14 shows the input voltage waveforms applied to different nodes when any given crypto engine is selected for encoding. The encryption process comprises photosensing, information encoding, and erase cycles. $V_{N1}$, $V_{N2}$, $V_{N6}$, and $V_{N7}$ are held constant at 1 V, 5 V, 0 V and 0 V, respectively, throughout the encryption process. $V_{N8}$ and $V_{N9}$ are enabled only during the encoding cycle, with 1 V and 5 V applied to the respective nodes. Finally, $V_{N4}$ cycles between $V_{illumin} = -4$ V, $V_{encoding} = 0$ V, and $V_{erase} = 11$ V during photosensing, encoding, and erase cycles, respectively. Note that $V_{N2} = 5$ V ensures that $T_{PST}$ is biased in the deep on-state with orders of magnitude lower resistance than $T_{PT}$ and $T_{WGNA}$. Therefore, the $V_{PSV}$ value obtained at node $N_5$ can be expressed using Eq. 1.

$$V_{PSV} = f(G_{PT}, G_{WGNA}) \qquad (1)$$

Here, $G_{PT}$, and $G_{WGNA}$ are, respectively, the conductance values of $T_{PT}$ and $T_{WGNA}$. Note that since both $T_{PT}$ and $T_{WGNA}$ are operated in their subthreshold regime, none of these memtransistors behave like a linear resistor, which is why the $V_{PSV}$ value cannot be derived based on the standard model of a resistor divider network. Nevertheless, the 64 $MoS_2$ memtransistors used as $T_{WGNA}$ corresponding to each of the $8 \times 8$ crypto engines are randomly pre-programmed such that the $V_{PSV}$ values obtained under dark condition follow a Gaussian distribution with a mean of 0.3 V and finite $\sigma_G$ when measured using a back-gate voltage, $V_{N6} = 0$ V (see Supplementary Fig. 15 for the distribution of $G_{WGNA}$ for different $\sigma_G$). While it can be argued that on-chip random number generators are more desirable solutions, these are often power-hungry and pose integration challenges. Instead, our approach offers inherent energy efficiency since the noise conductance values are already stored in $T_{WGNA}$, reducing the energy burden during field operations although it adds area and storage overhead. Similarly, the 64 $MoS_2$ memtransistors used as $T_{PT}$ corresponding to each of the $8 \times 8$ crypto engines are pre-programmed such that the pre-illumination $G_{PT} = 10$ nS when measured using a back-gate voltage, $V_{N2} = 0$ V. To transcribe the LED illumination, $T_{PT}$ is biased at $V_{N2} = V_{illumin} = -2$ V for $t_{illumin} = 100$ ms. If the pixel illumination is present ($I_{LED} = 20$ mA), the corresponding $T_{PT}$ reaches $G_{PT} \approx 400$ nS when measured post-illumination using $V_{N2} = 0$ V. Figure 3a shows the $V_{PSV}$ values corresponding to each of the 64 crypto engines and Fig. 3b shows the corresponding colormaps for $V_{PSV}$ for different values of $\sigma$ when an $8 \times 8$ pixelated image of the letter 'N' is transduced using the corresponding $T_{PT}$ and $T_{WGNA}$. Under no-noise conditions, i.e., $\sigma_G = 0$, $V_{PSV}$ values, as expected, achieve two distinct levels, ~0.3 V and ~0.6 V, corresponding to dark and bright pixels in the image of the letter 'N' and the colormap of $V_{PSV}$ shows accurate transduction of the optical information. For any finite $\sigma_G$, both $V_{PSV}$ levels become superimposed with random Gaussian noise. Higher $\sigma_G$ values are translated into more variation in the two $V_{PSV}$ levels in Fig. 3a and the corresponding colormaps in Fig. 3b, obscuring the sensed information.

The final stage of the encryption process is performed by the $MoS_2$ memtransistors used as $T_{SN}$, which transforms the $V_{PSV}$

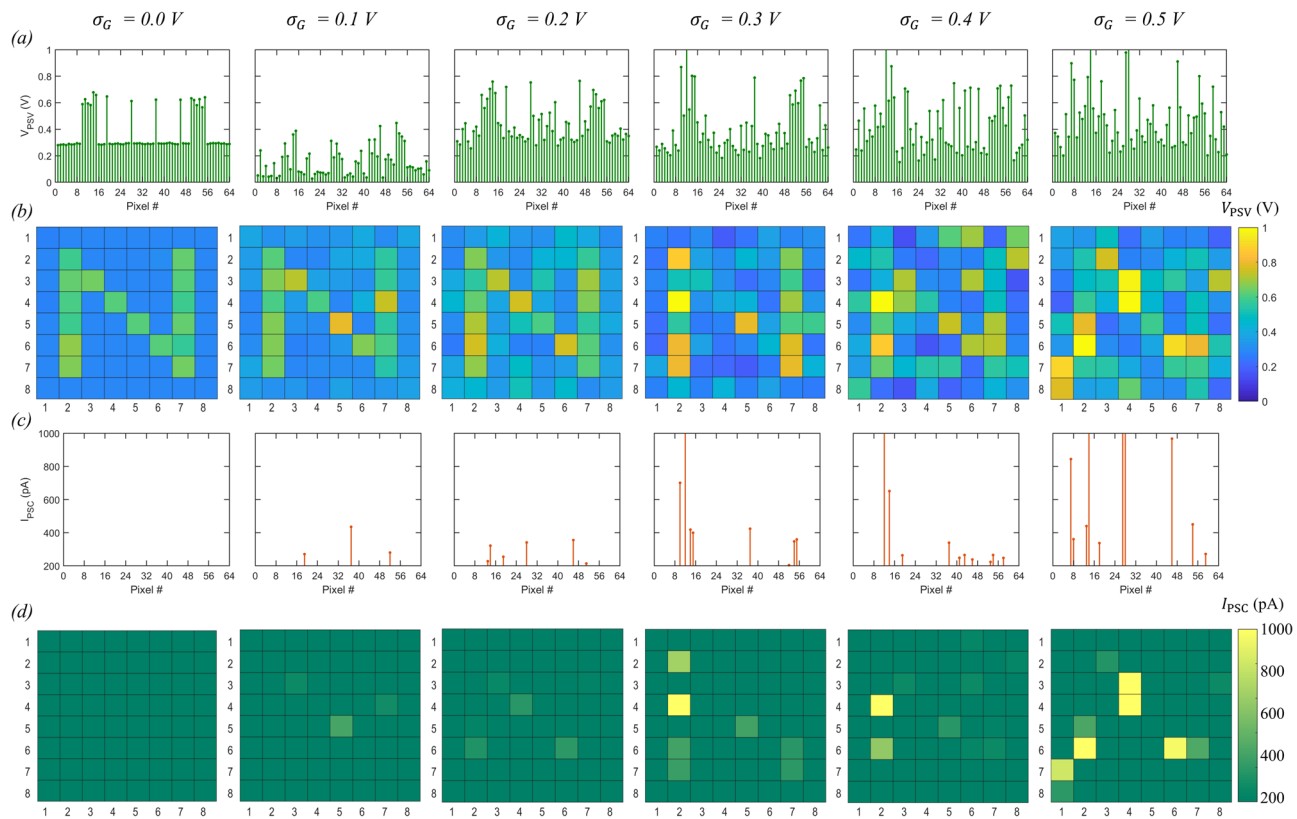

**Fig. 3 Encryption process. a** Pre-synaptic voltage, $V_{PSV}$. **b** Colormaps of $V_{PSV}$. **c** Post-synaptic current, $I_{PSC}$. **d** Colormaps of $I_{PSC}$ for different values of noise standard deviation, $\sigma_G$, when an 8 × 8 pixelated image of the letter 'N' is sensed and transduced using the corresponding $T_{PT}$, $T_{WGNA}$, and $T_{SN}$.

into a corresponding $I_{PSC}$. Note that $V_{N9} = 5$ V ensures that $T_{EST}$ is biased in the deep on-state with orders of magnitude lower resistance than $T_{SN}$, allowing $I_{PSC}$ to be determined by $T_{SN}$ and hence $V_{PSV}$. Also, note that all 64 $MoS_2$ memtransistors used as $T_{SN}$ are pre-programmed such that their encoding threshold is at $V_{ET} = 0.5$ V (see Supplementary Fig. 16 for the transfer characteristics of a representative $T_{SN}$). This ensures that the $V_{PSV}$ values obtained in Fig. 3a are primarily subthreshold with occasional threshold crossing events. In other words, for $V_{PSV} < V_{ET}$, $T_{SN}$ does not invoke any observable current response greater than the noise floor of the measurement (~1 pA/μm). Figure 3c shows the $I_{PSC}$ values obtained from $T_{SN}$ corresponding to each of the 8 × 8 crypto engines and Fig. 3d shows the corresponding colormaps for $I_{PSC}$ for different $\sigma_G$ for the letter 'N'. As evident, for lower $\sigma_G$ values, there are few if any threshold crossing events, resulting in sporadic bright pixels in the 8 × 8 encrypted image of the letter 'N', whereas for higher $\sigma_G$ values, there are more frequent threshold crossing events, resulting in the greater number of bright pixels in the 8 × 8 encrypted image of the letter 'N'. Nevertheless, the $I_{PSC}$ map constitute the encoded information for the letter 'N'. Note that, except for $T_{PT}$, none of the other $MoS_2$ memtransistors corresponding to the crypto engine are biased in their deep off-state and hence these memtransistors do not encounter any photogating effect.

**Encryption strenth of encoded information.** To analyze the strength of the encryption process, we define true positive (TP) as an event when a bright pixel in the encoded image corresponds to a bright pixel in the original image and false positive (FP) as an event when a bright pixel in the encoded image corresponds to a dark pixel in the original image. The likelihood of identifying the letter 'N' by an eavesdropper from the encrypted image will, therefore, be determined by the detectivity (D), which is defined

as $D = p_{TP} - p_{FP}$, where $p_{TP}$ is the probability of TP and $p_{FP}$ is the probability of FP. Colormaps in Fig. 4a–c, respectively, show $p_{TP}$, $p_{FP}$, and D as a function of $\sigma_G$ obtained by repeating the experiments $P = 50$ times, which represents the population strength, and Fig. 4d shows the corresponding population means. Note that the population means for D exhibit a non-monotonic behavior. At a low noise level, there is hardly any FP, i.e., low $p_{FP}$, but the likelihood of detecting the letter 'N' remains low due to limited threshold crossing events for the original bright pixels, i.e., low $p_{TP}$. At a high noise level, both bright and dark pixels corresponding to the original image cross the spiking threshold, resulting in high $p_{TP}$ and $p_{FP}$ and, therefore, low D. However, at an intermediate noise, the detectivity reaches its maximum value. Figure 4e shows the number of brute force trials (BFTs) by the eavesdropper necessary to identify the letter 'N' as a function of $\sigma_G$. Note that we computed $BFT = 1/D^S$, where $S = 8 \times 8 = 64$ is the size of the image. The number of BFTs is found to be significantly high irrespective of $\sigma_G$. Furthermore, the number of BFTs increases exponentially with $S$ (see Supplementary Fig. 17). Therefore, the encryption can be considered to be secure from an eavesdropper with finite resources. Figure 4f shows the average energy expenditure by the crypto engines ($E_{encrypt}$) for the encryption of the letter 'N" as a function of $\sigma_G$ calculated using Eq. 2.

$$E_{encrypt} = \frac{1}{64} \sum_{i=1}^{64} [I_{N1N7,i} V_{N1} + + I_{N8N7,i} V_{N8}] t_{illumin} \\ + 1/2 C_G (V_{N2}^2 + V_{N4}^2 + V_{N6}^2 + V_{N9}^2] \tag{2}$$

Here, $I_{N1N7,i}$ and $I_{N8N7,i}$ are the currents flowing through the sensing and encoding units, respectively. Note that the energy expenditure was found to be less than a few hundred picojoules per engine even for the highest $\sigma_G$. See Supplementary

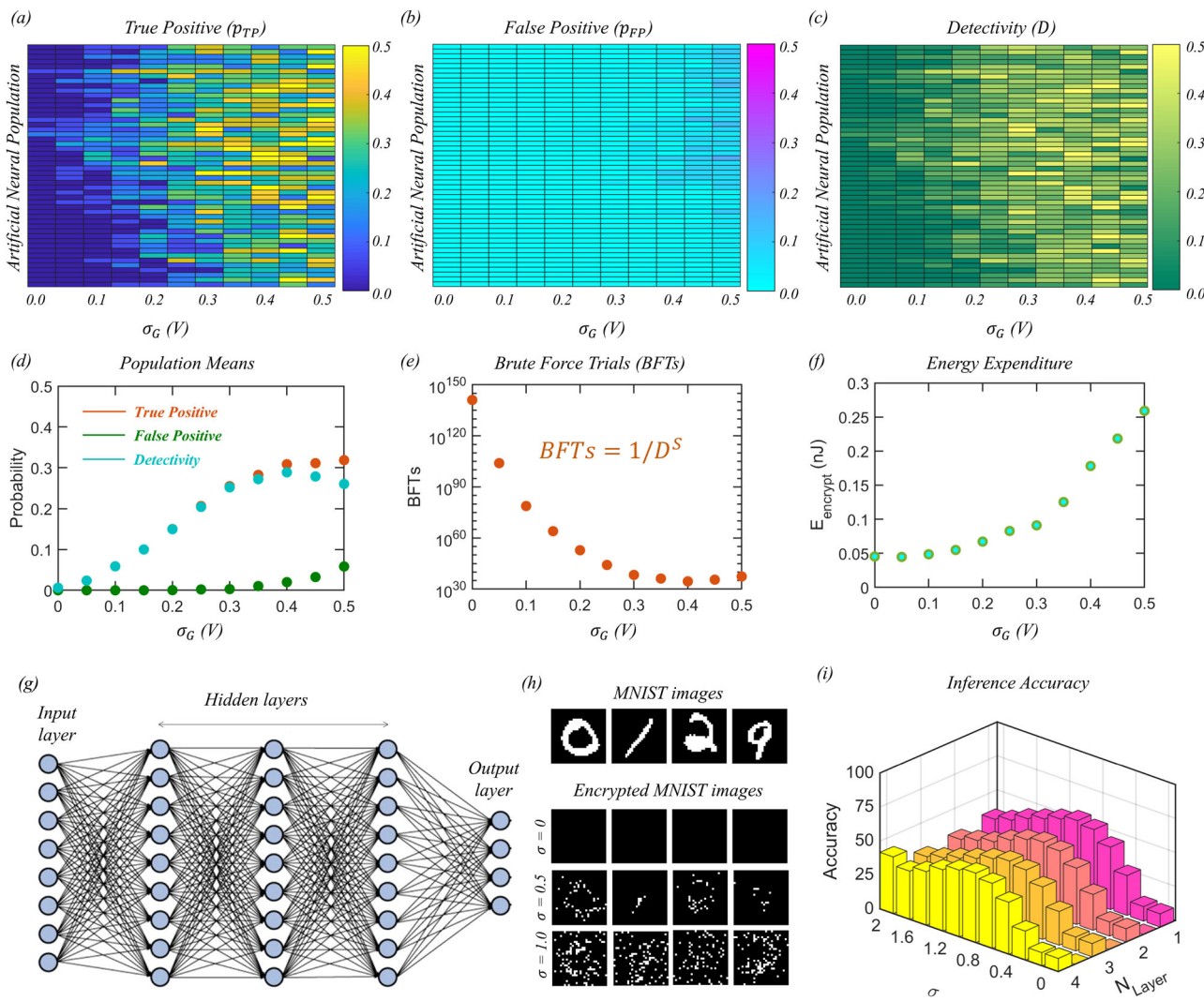

**Fig. 4 Encryption strength.** Colormaps of the likelihood, or probability, of (**a**) true positive ($p_{TP}$), (**b**) false positive ($p_{FP}$), and (**c**) detectivity ($D = p_{TP} - p_{FP}$) as a function of $\sigma_G$ for $P = 50$ encoders. **d** Corresponding population means. True positive (TP) is an event when a bright pixel in the encoded image corresponds to a bright pixel in the original image, and false positive (FP) is an event when a bright pixel in the encoded image corresponds to a dark pixel in the original image. **e** The number of brute force trials (BFTs) by the eavesdropper necessary to identify the letter 'N' as a function of $\sigma_G$. Note that BFT = $1/D^S$, where $S = 8 \times 8 = 64$ is the size of the image. **f** The average energy expenditure for the encryption process as a function of $\sigma_G$. **g** A deep neural network (DNN) trained to recognize the MNIST data set for digit classification. The training and testing sets consisted of 60,000 and 10,000 images, respectively. **h** Representative MNIST images with white Gaussian noise (WGN) of different standard deviations ($\sigma$) binarized at a threshold of 1.5, mimicking our MoS$_2$ memtransistor-based encryption process. **i** Average inference accuracy for 10,000 encrypted images as a function of $\sigma$ and the number of hidden layers ($N_{Layer}$). Irrespective of $\sigma$, the inference accuracy remains low, indicating the robustness of our bio-inspired encryption to trained DNNs.

Note 1 for the energy expenditure comparison with other competing technologies.

The encryption strength is also tested assuming that the eavesdropper has access to a trained artificial DNN, with the information being communicated being an encrypted MNIST data set for digit classification. DNNs have shown remarkable success in various applications ranging from image recognition and pattern classification to defeating professional players in the game of "Go"[63]. DNNs consist of one input layer, one output layer, and many hidden layers ($N$), with each containing a certain number of neurons. The neurons in a layer and its consecutive layer are connected by synapses and the strength of the synapses, i.e., synaptic weights, are trained using back-propagation algorithms. The deeper the network is, i.e., for higher $N$, the more complex datasets that can be trained, and with better accuracy. Figure 4g shows the schematic of a DNN trained to recognize the MNIST

data set. The input layer consists of 784 neurons where the normalized pixel values of the images ($28 \times 28$ pixels) in the MNIST dataset are sent and the output layer consists of 10 neurons that correspond to digits from 0 to 9. Output neurons perform the SoftMax activation function, and the winner is determined based on the maximum value. We have used $N_{Layer} = 1, 2, 3, 4$ hidden layers with 300, 200, 100, and 50 neurons in the corresponding layers, respectively. The gradient descent algorithm is used to train the DNN using 60,000 images with a learning rate of 0.003 and rectified linear unit (ReLU) as the activation function, and the remaining 10,000 images are used for testing the inference accuracy. Supplementary Fig. 18 shows the training and inference accuracy as a function of $N_{Layer}$ and the number of epochs. Training accuracies of 100% and testing accuracies of >98% were achieved beyond 50 epochs irrespective of $N_{Layer}$. Note that higher $N_{Layer}$ values achieve better training

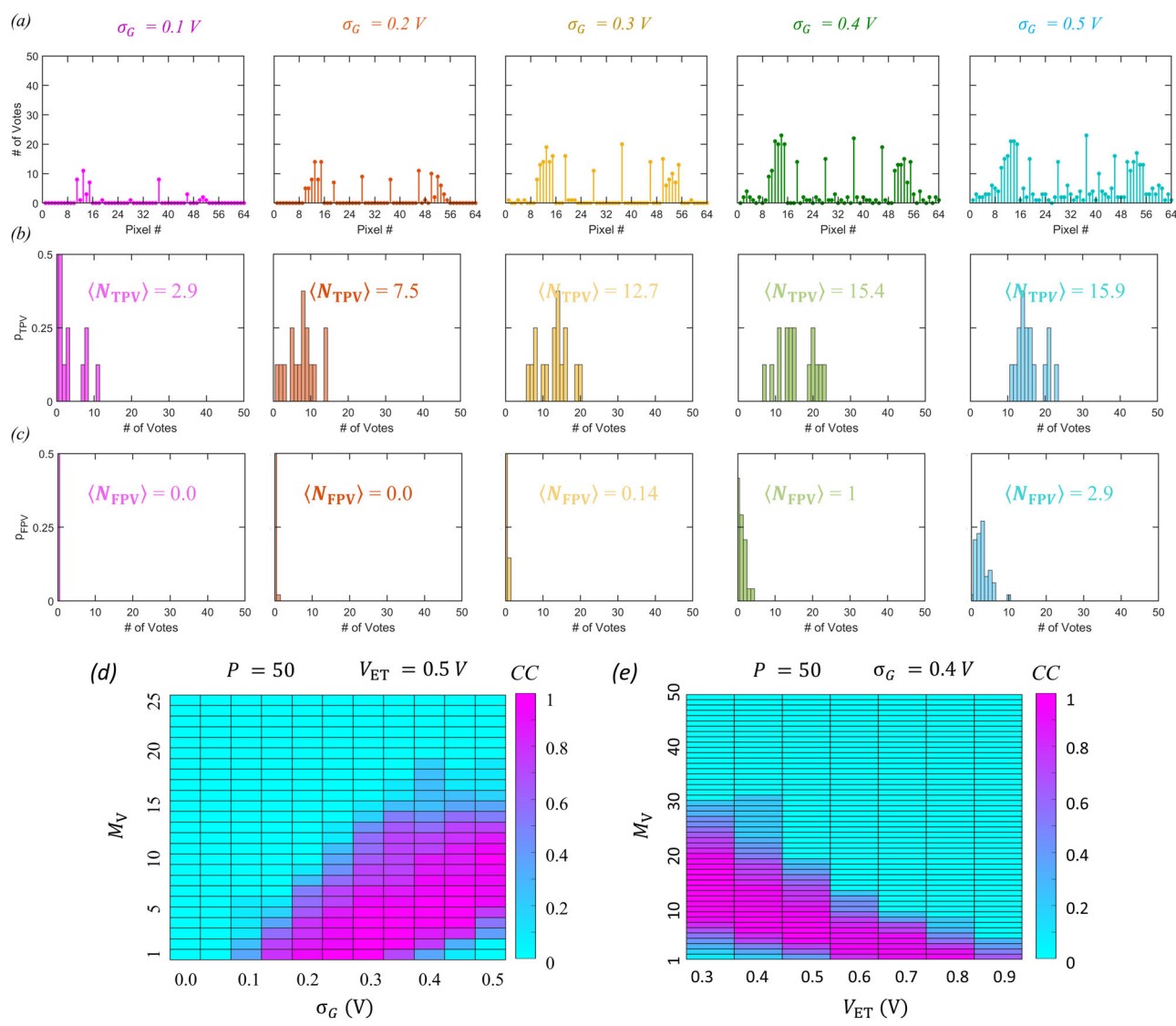

**Fig. 5 Voting-based decryption process. a** Number of votes corresponding to each pixel of the encoded images of the letter 'N' received from $P = 50$ encoders for different $\sigma_G$. A vote is registered when the encoded pixel is bright, i.e., $I_{PSC} > 350$ pA. The vote is a true positive vote (TPV) if the corresponding pixel in the original image is also bright, whereas the vote is a false positive vote (FPV) if the corresponding pixel in the original image is dark. Probability distribution for (**b**) TPVs ($p_{TPV}$) and (**c**) FPVs ($p_{FPV}$) corresponding to (**a**). Insets show the expected number of TPV, i.e., $\langle N_{TPV} \rangle = \sum_{n=1}^{P} n p_{TPV}(n)$, and FPV, i.e., $\langle N_{FPV} \rangle = \sum_{n=1}^{P} n p_{FPV}(n)$. **d** The colormap of the correlation coefficient (CC) between the original and the decrypted image images of the letter 'N' as a function of $\sigma_G$ and $M_V$ when encryption is done by $P = 50$ encoders with encoding thresholds of $V_{ET} = 0.5$ V. Here, $M_V$ is the minimum number of votes required to mark a pixel as bright. **e** The colormap of CC between the original and decrypted images of the letter 'N' as a function of $V_{ET}$ and $M_V$ for $\sigma_G = 0.4$ V and $P = 50$. Without prior knowledge of $\sigma_G$, $P$, and $V_{ET}$ it is difficult to determine $M_V$ and hence decode the information.

and inference accuracies with a lesser number of epochs. Next, we added white Gaussian noise (WGN) to 10,000 MNIST testing images and binarized them at a threshold of 1.5, mimicking the encryption process of our MoS$_2$ memtransistor-based crypto engines. Figure 4h shows some examples of encoded MNIST images for different standard deviations ($\sigma$) of the WGN. Figure 4i shows the inference accuracy for the encrypted images as a function of $\sigma$ for different $N_{Layer}$. Interestingly, the accuracy values are found to be significantly low irrespective of $\sigma$ and $N_{Layer}$, indicating the robustness of our proposed encryption scheme to trained DNNs.

**Voting-based decryption process.** In order to retrieve the information, we adopt a population voting-based algorithm. We

assume that the encoded images of the letter 'N' are transmitted over different communication channels by $P$ encoders, each comprising $8 \times 8$ array of the crypto engines. The receiver at the other end receives $P$ encoded images and counts the number of votes corresponding to each pixel. In our experiments, this was accomplished by encoding the same information $P$ times by the same $8 \times 8$ array of the crypto engines but using different $G_{WGNA}$ every time. Figure 5a shows the vote counts for each pixel when $P = 50$ for different $\sigma_G$. A vote is registered when the encoded pixel is bright, i.e., $I_{PSC} > 350$ pA. The vote is considered to be a true positive vote (TPV) if the corresponding pixel in the original image is also bright, whereas the vote is considered to be a false positive vote (FPV) if the corresponding pixel in the original image is dark. Figure 5b, c, respectively, show the probability distribution for TPVs ($p_{TPV}$) and FPVs ($p_{FPV}$) for $P = 50$ for

different $\sigma_G$. At low noise levels, the probability of crossing the encoding threshold ($V_{ET}$) is low and hence only a few encoders fire simultaneously, resulting in lower expected number of encoders for TPV, i.e., $\langle N_{TPV} \rangle = \sum_{n=1}^{P} n p_{TPV}(n)$. The expected number of encoders for FPV, i.e., $\langle N_{FPV} \rangle = \sum_{n=1}^{P} n p_{FPV}(n)$, is even lower. Similarly, at high noise levels, the probability of crossing the threshold of the encoder is high and hence more encoders fire synchronously, resulting in larger $\langle N_{TPV} \rangle$ and $\langle N_{FPV} \rangle$. However, as seen in Fig. 5b, c, for any $\sigma_G$, $\langle N_{TPV} \rangle$ is higher than $\langle N_{FPV} \rangle$. Supplementary Fig. 19 shows the decoding of the images of the letter 'N' for different $\sigma_G$ for different number of mandated votes ($M_V$) to mark a pixel as bright for $P = 50$. Figure 5d shows the corresponding colormap of the correlation coefficient ($CC$) between the original and the decrypted image as a function of $\sigma_G$ and $M_V$. Note that, for a given $\sigma_G$, there is an optimum range for $M_V$, that allows accurate decryption of the encoded image, i.e., $CC = 1$. Supplementary Fig. 20 shows similar results for $CC$ when different encoding population sizes ($P$) are used. As expected, the optimum number of $M_V$ for accurate decryption is found to be different for similar $\sigma_G$. Therefore, without the prior knowledge of the $\sigma_G$ and $P$ used by the bio-mimetic encoder, it is difficult to decode the information.

The strength of encoding can be further enhanced by exploiting the programming capability of our MoS$_2$ memtransistors. Here we reconfigure the encoding threshold ($V_{ET}$) in a manner similar to neuroplasticity in biological neurons allowing adaptation to changing environments and stimuli. Supplementary Fig. 21 shows the encryption of the letter 'N' by encoders with different $V_{ET}$ for different $\sigma_G$. As obvious, if $V_{PSV} > V_{ET}$, the encryption process is pointless, or, in other words, the communication is insecure. For $V_{ET}$ values slightly greater than $V_{PSV}$, there are more threshold crossing events even for low $\sigma_G$, whereas for $V_{ET}$ values further from $V_{PSV}$, there are limited threshold crossing events even for high $\sigma_G$. Figure 5e shows the colormap of $CC$ between the original and decrypted images of the letter 'N' as a function of $V_{ET}$ and $M_V$ for $\sigma_G = 0.8$ V and $P = 50$ (see Supplementary Fig. 22 for similar results with different $\sigma_G$). Clearly, the optimum $M_V$ for accurate decryption is found to be different for different $V_{ET}$. Therefore, not only $\sigma_G$ and $P$, but also prior knowledge of $V_{ET}$ is required for decoding the information, which makes the system more robust from the eavesdropper. However, to achieve on-field reconfiguration capability, some peripheral circuits will be necessary for our integrated crypto engines, for example, the clocking signal for the MoS$_2$ memtransistor based $T_{PT}$, the gate-voltages necessary to randomly program the MoS$_2$ memtransistor based $T_{WGNA}$, and the gate voltages necessary to program the encoding threshold for the MoS$_2$ memtransistor based $T_{SN}$. Finally, the reader may be interested in knowing the impact of environmental noise, temperature, etc., on the encryption process. We have found that the device-to-device variation is dominant compared to environmental noise and that it has insignificant impact on the encryption process. Similarly, small variations in temperature are expected to have minimal impact on the proposed crypto engine.

## Discussion

In conclusion, we have experimentally demonstrated a monolithically integrated and multi-pixel *all-in-one* hardware IoT platform based on programmable and multifunctional MoS$_2$ memtransistors which is capable of sensing, storing, and securing information. Since a single material and similar device structures are used for our demonstration, the hardware investment is minimal. The average energy expenditure by each crypto engine was also found to be low, in the range of a few hundred picojoules, as we have primarily exploited the subthreshold regime of memtransistor operation. The encrypted information is found to be secure from an adversary with access to advanced machine learning capabilities such as trained DNNs. The decryption of the encrypted information relies on population voting. Overall, our platform shows the capability of near-sensor security by exploiting in-memory bio-inspired computing.

## Methods

**Fabrication of local back-gate islands.** The local back-gate islands were fabricated on a commercially-bought substrate (285 nm SiO$_2$ on p$^{++}$-Si). The substrate was spin-coated with a bilayer photoresist consisting of Lift-Off-Resist (LOR 5A) and Series Photoresist (SPR 3012), which were then baked at 185 °C for 2 minutes and 95 °C for 1 minute, respectively. The bilayer photoresist was then exposed using a Heidelburg Maskless Aligner (MLA 150) to define the islands and developed using MF CD26 microposit, followed by a de-ionized (DI) water rinse for 75 and 60 seconds respectively. The local back-gate island electrodes (20/50 nm TiN/Pt) were deposited using reactive sputtering. The photoresist was then removed using acetone and Photo Resist Stripper (PRS 3000) and cleaned using 2-propanol (IPA) and DI water. An atomic layer deposition (ALD) process was then implemented to grow 50 nm Al$_2$O$_3$ uniformly across the entire substrate, including the island regions. To access the individual Pt back-gate electrodes, etch patterns were defined using the same bilayer photoresist (LOR 5A and SPR 3012) used previously. The bilayer photoresist was then again exposed using the MLA 150 and developed using MF CD26 microposit. The 50 nm Al$_2$O$_3$ was subsequently dry etched using a BCl$_3$ reactive ion etch chemistry at 5 °C for a total of 80 s; this process was split into four 20 s etches to minimize heating in the substrate and thus ensure a uniform etch rate/depth. The photoresist was then removed to give access to the individual Pt electrodes.

**Monolayer MoS$_2$ Film Growth.** Monolayer MoS$_2$ was obtained from the Pennsylvania State University 2D Crystal Consortium (2DCC)[64]. It was deposited on an epi-ready 2" c-sapphire substrate by metal-organic chemical vapor deposition (MOCVD). An inductively heated graphite susceptor equipped with wafer rotation in a cold-wall horizontal reactor was used to achieve uniform monolayer deposition as previously described[65]. Molybdenum hexacarbonyl (Mo(CO)$_6$) and hydrogen sulfide (H$_2$S) were used as the precursors. Mo(CO)$_6$ was maintained at 10 °C and 950 Torr in a stainless-steel bubbler, which was used to deliver 0.036 sccm of the metal precursor for the growth, while 400 sccm of H$_2$S was used. MoS$_2$ deposition was carried out at 1000 °C and 50 Torr in H$_2$ ambient, and monolayer growth was achieved in 18 min. The substrate was first heated to 1000 °C in H$_2$ and maintained for 10 min before the growth was initiated. After growth the substrate was cooled in H$_2$S to 300 °C to inhibit decomposition of the MoS$_2$ films.

**MoS$_2$ film transfer to local back-gate islands.** To fabricate the MoS$_2$ memtransistors, the as-grown monolayer MoS$_2$ film was transferred from the sapphire growth substrate to the SiO$_2$/p$^{++}$-Si substrate with local back-gate islands using a PMMA (polymethyl-methacrylate) assisted wet transfer process. PMMA 495 A6 resist was spun onto the growth substrate at 4000 RPM for 45 s and allowed to sit overnight to ensure good PMMA/MoS$_2$ adhesion. The edges of the spin-coated film were then scratched using a razor blade and the substrate was immersed into a 2 M NaOH solution kept at 90 °C. Capillary action served to draw the NaOH solution to the PMMA/substrate interface, separating the hydrophobic PMMA/MoS$_2$ from the hydrophilic sapphire substrate. Note that scratching the edges of the film served to aid this process *via* removing any PMMA beading that may have been formed at the edge of the substrate upon spinning and shortening the distance for the solution to penetrate. The detached film was retrieved from the NaOH bath using a clean glass slide and rinsed three times in separate DI water baths (15 min each). It was then retrieved from the final bath using the prepared SiO$_2$/p$^{++}$-Si substrate with local back-gate islands and baked at 50 °C and 70 °C for 10 min each to remove moisture and promote adhesion. Finally, the PMMA supporting layer was removed using an acetone bath and the substrate was cleaned using IPA.

**Fabrication of monolayer MoS$_2$ memtransistor.** To define the channel regions for the memtransistors, the sample was first spin-coated with PMMA 950 A6 at 4000 RPM for 45 s and then baked at 180 °C for 90 s. Electron-beam (e-beam) lithography was used to pattern the resist, which was developed using a 1:1 mixture of 4-methyl-2-pentanone (MIBK) and IPA for 60 s and pure IPA for 45 s. The defined channels were separated *via* dry etching using a sulfur hexafluoride (SF$_6$) reactive ion etch chemistry at 5 °C for 30 s. Following the etch step, the sample was rinsed in acetone for 30 min to remove the remaining photoresist, followed by an IPA bath to clean the sample. To define the source and drain contacts, sample was spin-coated with methyl methacrylate (MMA) and baked at 150 °C for 90 s before applying PMMA A3, which was baked 185 °C for 90 s. Both resists were spun at 4000 RPM for 45 s. E-beam lithography was used to pattern the source and drain contacts, and development was again performed using a 1:1 mixture of MIBK and

IPA for 60 s and pure IPA for 45 s. Note that this development process allowed for the formation of a significant undercut in the bilayer resist, making subsequent metal deposition/liftoff easy. 40 nm of nickel (Ni) and 30 nm of gold (Au) were deposited using e-beam evaporation. Finally, lift-off of the evaporated material was performed by immersing the sample in acetone for 30 min and in IPA for 15 min. In the final design, each local back-gate island contained one memtransistor to allow for individual gate control.

**Monolithic integration**. Each crypto engine of our multi-pixel ($8 \times 8$) hardware consists of 5 MoS$_2$ memtransistors, as shown using the circuit schematic in Fig. 1d. To define the connections between the respective nodes of $T_{PST}$, $T_{PT}$, $T_{WGNA}$, $T_{EST}$, and $T_{SN}$, the substrate was spin-coated with MMA and PMMA, followed by e-beam lithography and development using a 1:1 mixture of MIBK and IPA, e-beam evaporation of 60 nm of Ni and 30 nm of Au, and, finally, rinsing away the resist using acetone and IPA. All column-select metal lines were defined during this lithography step. The row-select lines were also defined except for the regions, where they crossover the column-select lines. To electrically isolate the column- and row-select lines, the substrate was again spin-coated with MMA and PMMA, followed by e-beam lithography and e-beam evaporation 80 nm of insulating alumina (Al$_2$O$_3$) at the cross-points. Finally, another lithography step was performed, followed by the deposition of 60 nm of Ni and 30 nm of Au to connect the row-select lines bridging over the cross-points.

**Electrical characterization**. Electrical characterization of the fabricated devices was performed using n automated probe station (Cascade SUMMIT200) under atmospheric conditions using a Keysight B1500A parameter analyzer. We have used our automated probe station for the readout, i.e., the probes move from one set of select lines to the next to collect the data. The entire encryption process is, therefore, automated without the need for any human intervention making the demonstration real-time. See Supplementary Video 1 for a representative recording.

## Data availability
Relevant data supporting the key findings of this study are available within the article and the Supplementary Information file. All raw data generated during the current study are available from the corresponding authors upon request.

## Code availability
The codes used for plotting the data are available from the corresponding authors on reasonable request.

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

## Acknowledgements
Authors would like to acknowledge Shiva Subbulakshmi Radhakrishnan for his help with the DNN simulation. The work of A.D. and S.D. was partially supported by Army Research Office (ARO) through Contract Number W911NF1920338 and National Science Foundation (NSF) through a CAREER Award under grant no. ECCS-2042154. Authors also acknowledge the materials support from the National Science Foundation (NSF) through the Pennsylvania State University 2D Crystal Consortium–Materials Innovation Platform (2DCCMIP) under NSF cooperative agreement DMR-2039351.

## Author contributions
S.D. conceived the idea and designed the experiments. S.D., and A.D. performed the experiments, analyzed the data, discussed the results, agreed on their implications. N.T., and J.M.R. synthesized the MOCVD grown monolayer MoS₂. All authors contributed to the preparation of the manuscript.

## Competing interests
The authors declare no competing interests.
