## [Peer Review File · Nature Communications]

All-in-one, bio-inspired, and low-power crypto engines for near sensor security based on two-dimensional memtransistorsREVIEWER COMMENTS

Reviewer #1 (Remarks to the Author):

This manuscript demonstrates photodetectors, current adders, and encoders using multifunctional MoS₂ FETs and achieves a low-power cryptosystem that is suitable for the internet of things (IoT) requirements.

The electronic characteristics (mobility, on/off ratio and on current) of the MoS₂ transistors are comparable with other MoS₂ grown by chemical vapor deposition (CVD). The demonstration of MoS₂ array using the grown MoS₂ shows great potential for future two-dimensional (2D) material-based signal pre-processing for IoT applications. The authors analyzed the performance of photodetectors, the programmable threshold voltage (V_{TH}) and the efficiency for pattern encryption comprehensively, which are the key components for the crypto engine.

Although this work demonstrates a crypto engine with sufficient evaluation on encryption and signal decoding, explanations for some fundamental mechanisms and experimental details are still required to avoid misunderstanding for readers.

Major comments:

1. The authors need to provide clear images and explanations for their MoS₂ FET array. (i) The positions (layout) of CA, WGNA and AN on the crypto engine chip shown in Fig. S1 are missing. (ii) The zoomed-in images of Fig. 2b are required to show the details of these arrays. (iii) The authors need to explain why the FET arrays (Fig. 2b) are separated far away and what the black squares in between of them represent?

2. The authors need to evaluate the yield of the array fabrication, especially the transfer step. They should explain how the wrinkles and cracks affect the device yield since the wrinkles are obvious in Fig. 2b.

3. It seems that the authors only use one photodetector transistor to measure the IPH with a 64*1 array of LED voltages. However, for real applications, they should use 64 different devices to measure the IPH, because the device variation may also affect the pattern 'N' and the encoded image.

4. In Fig. 3a, it seems that the CA or WGNA is a physical adder that relies on Kirchhoff's law, indicating that at least three probes should be connected to the adder to connect two input terminals (IPH and Inoise) and one output terminals (Iout or VPSV) simultaneously. However, in Fig. S1, only two probes are connected to the crypto engine which is not consistent.

5. In Fig. S1, the LED can shine lights on the detector but also the encoder. The authors need to explain how the light from LED affects the noise current in the WGNA.

6. A key piece of information is missing that is how the authors store the measured IPH and Inoise (64*1 arrays). If they sum the I_{IPH} and Inoise directly after measurement, please go back to Q4 and 5. If they first measure the current and store them and use the measured current as inputs to the CA to get encoded output current, they actually use external memories to store the temporal information for next-step signal processing, which is not a fully MoS₂-based crypto engine. In this case, they need to clearly mention this data storage step.

7. The explanation about the floating gate (FG) is not convincing. (i) If the authors use heavily doped p++ silicon as the common back gate, the contact between TiN and Si should be ohmic contact since the high doping concentration already makes tunneling happen. (ii) Even if we assume the existence of the Schottky barrier (SB) at the interface, the forward direction for holes in this SB is from Si to TiN, which means the trapped negative charges can go through the SB easily (from TiN to Si). In this case, the SB cannot store the trapped negative charges in the TiN layer.

8. The x-axis (time) of Fig. 2f and Fig. 2h are missing, which are important for retention results. The authors need to show the time interval between the measurements after the VP pulse.

9. Fig. S4 is not convincing enough. (i) The authors need to mention the Vp pulse used here and the duration. (ii) The difference between Fig. S5a and S5b is not as obvious as that shown in the main text. Here, the Vth shift is less than 0.5 V. However, in Fig. 2f, the Vth shift can reach 2 V. The author should use the extreme condition (the largest Vth shift to the programmed device) to prove the negligible effect to unselected devices. (iii) The authors need to briefly explain the reason why there is no crosstalk effect to unselected devices.

10. The brute force trials (BFT) calculated here are not appropriate to evaluate the efficiency of the crypto engine. The information is an 8*8 'N' image, where pixels have interconnections and hidden information. To decode the information of this image, we do not need to exactly get all the pixels information correctly. The author should not emphasize the 'astronomical value'. The simulation of artificial neural networks (ANN) here is more reliable.

Minor comments:

1. During measurement, the authors need to clearly mention how they measure the current for the 64 noise transistors. They choose to either (i) program all the devices first and measure them or (ii) program one device and measure immediately and move to the second one.

2. Unit is missing (1.5 V) in the last sentence in Page 13.

Reviewer #2 (Remarks to the Author):

The main expertise of the authors is in the development of MoS2 FETs. In this paper, they experiment and develop an IOT edge sensor. The authors claim that this new design provides encryption.

The main problem with the paper is the claimed security level. The first 6 references are related to security, the rest is related to MoS2 FET devices and designs.

These first 6 references are absolutely not state of the art. RSA is never used in sensor nodes. Much more low power and low energy solutions exist, for instance based on elliptic curves. Public key crypto systems that fit in the power budget of passive RFIDs have been developed.

The proposed solution only provides encryption, i.e. hiding of the information. The solution still cannot address what the authors call "loss, misuse and manipulation" in the introduction. It seems that the attacker is only allowed 'passive eavesdropping' meaning that he can only observe the images that pass by.

Figure 4, subfigure (d), shows the population means and the detectivity. This detectivity reaches its maximum of around 45% for a σ_V around 0.75.

At this σ_V , subfigure (e) shows that the attacker needs around 10^{20} Brute Force Trials. This corresponds to 66 bits of security. Modern IOT devices require 128 bits of security for brute force attacks.

Reviewer #3 (Remarks to the Author):

The authors demonstrate an all-in-one biomimetic IoT hardware platform based on multifunctional MoS₂ FETs that integrates sensing, non-volatile storage, and security. The sensing elements are realized in a MoS₂ field effect transistor-technology with a programmable gate stack that can be used as a photodetector (PD). The term biomimetic stems from the analogy of the cryptoengine to an artificial neural encoder.

In principle the mechanism of sensing, encoding and decoding is explained well, but some information especially on transistor and circuit level is missing. For example, both the WGNA and the AN are realized with thin film MoS₂ transistors (in form of a current adder and look up table as explained later), but what is the exact schematic? Also, is there a difference in terms of layout, physical realization, or are all MoS₂ transistors identical? How is the integration done between WGNA and AN? Are they connected to independent voltage sources?

Also, is the generation of white noise really guaranteed for MoS₂? Many thin film transistors based on organic and inorganic channel materials show $1/f$ or even $1/f^\gamma$ noise for low frequencies. What is the typical delay of the used MoS₂ transistors (e.g. obtained from pulsed measurements or by creating inverter cells)?

I believe the significance of this work is solid, however there are many works on optical artificial neural networks and neuromorphic computing architectures, that could be competitors for the artificial encoding presented here with possibly higher information complexity.

Just a few references, which are by far not comprehensive:

Ref. 1: Feldmann, J., Youngblood, N., Wright, C.D. et al. All-optical spiking neurosynaptic networks with self-learning capabilities. *Nature* 569, 208–214 (2019). <https://doi.org/10.1038/s41586-019-1157-8>

Ref. 2:

Lei Yin, Cheng Han, Qingtian Zhang, Zhenyi Ni, Shuangyi Zhao, Kun Wang, Dongsheng Li, Mingsheng Xu, Huaqiang Wu, Xiaodong Pi, Deren Yang, Synaptic silicon-nanocrystal phototransistors for neuromorphic computing, *Nano Energy* 63 (2019), doi.org/10.1016/j.nanoen.2019.103859.

The quality of data and especially the statistical analysis appears very reasonable and sound.

It would be nice to have some evidence for the robustness of the chosen design, meaning figure of merits such as reliability w.r.t. to typical environmental noise, temperature etc. However, these values need extensive characterization data and if not available, could maybe be simply commented based on the authors experience with the technology in the text.

The manuscript clearly states the context and gives explanations, however the adjective "biomimetic" in the title I personally would have omitted since it obscures the term "artificial neuron based on MoS₂ FETs" to form an AN encoder. So I would recommend the term "artificial neural encoder" instead but leave it up to the authors to decide.

References, are modest, out of 21 ~6 are self-citations, so the list could be extended to include also other possibilities of IOT security, such as memory less systems e.g. with physical unclonable functions (PUFs) and optical ANs, also some more IoT security and neuromorphic computing systems.

e.g. M. Z. Alom and T. M. Taha, "Network intrusion detection for cyber security on neuromorphic computing system," 2017 International Joint Conference on Neural Networks (IJCNN), Anchorage, AK, 2017, pp. 3830-3837, doi: 10.1109/IJCNN.2017.7966339.

REVIEWER COMMENTS

Reviewer #1 (Remarks to the Author):

This manuscript demonstrates photodetectors, current adders, and encoders using multifunctional MoS₂ FETs and achieves a low-power cryptosystem that is suitable for the internet of things (IoT) requirements.

The electronic characteristics (mobility, on/off ratio and on current) of the MoS₂ transistors are comparable with other MoS₂ grown by chemical vapor deposition (CVD). The demonstration of MoS₂ array using the grown MoS₂ shows great potential for future two-dimensional (2D) material-based signal pre-processing for IoT applications. The authors analyzed the performance of photodetectors, the programmable threshold voltage (V_{TH}) and the efficiency for pattern encryption comprehensively, which are the key components for the crypto engine.

Although this work demonstrates a crypto engine with sufficient evaluation on encryption and signal decoding, explanations for some fundamental mechanisms and experimental details are still required to avoid misunderstanding for readers.

We would like to thank the reviewer for appreciating our efforts in demonstrating the crypto engine. Following reviewer's recommendations, we have now performed additional experiments and included detailed description of the experimental procedures to avoid misunderstandings for the readers. In fact, we have now realized an 8×8 array of the crypto engines to encode 8×8 -pixel images with each crypto engine integrating five locally gated transistors (5T cell). Our redesigned hardware platform utilizes a total of 320 MoS₂ FETs.

Major comments:

1. The authors need to provide clear images and explanations for their MoS₂ FET array. (i) The positions (layout) of CA, WGNA and AN on the crypto engine chip shown in Fig. S1 are missing. (ii) The zoomed-in images of Fig. 2b are required to show the details of these arrays. (iii) The authors need to explain why the FET arrays (Fig. 2b) are separated far away and what the black squares in between of them represent?

We agree with the reviewer and apologize for the unclear images. We have now provided the high-magnification optical images of the fully integrated 8×8 crossbar array of the crypto engines to

encode 8×8 -pixel images with each crypto engine integrating five locally gated transistors (5T cell). Our entire hardware platform utilizes a total of 320 MoS₂ FETs.

Fig. R1 and Fig. R2, respectively, show the optical images of the entire chip and the 8×8 array of the crypto engines. Fig. R3a-b, respectively, show the optical images of each crypto engine with five MoS₂ FETs (5T cell) that integrates sensing, storage, and encoding functionalities, and of individual MoS₂ FETs, which are locally back-gated using a stack comprising of atomic layer deposition (ALD) grown 50 nm Al₂O₃ on sputter deposited 50/20 nm Pt/TiN. All back-gate islands were placed on a commercially purchased SiO₂/p⁺⁺-Si substrate. The Al₂O₃/Pt/TiN gate islands

Figure R1. Optical image of the all-in-one bio-inspired crypto chip.

not only allow non-volatile programming of our MoS₂ FETs but also enhances the photoresponse of MoS₂ FETs owing to the phenomenon of gate-tunable persistent photoconductivity when subjected to right polarity and magnitude of local back-gate biases. This, in turn, empowers our hardware platform to enable in-memory computing and near-sensor security, which are presently lacking for the conventional silicon technology as well as emerging solutions based on memristive

crossbar architectures. Fig. R3c-d, respectively, show the circuit schematic for the crossbar architecture and each crypto engine. In short, MoS₂ FETs used as the photo transistor (T_{PT}) mimic sensory neurons and transduce optical information into persistent photoconductance (G_{PT}), MoS₂ FETs used as the white Gaussian noise adder (T_{WGNA}) emulate noisy synapses and are pre-programmed into random conductance states (G_{WGNA}) to superimpose white Gaussian noise of

Figure R2. Optical image of the crossbar array of the crypto engines. Note that several (six) optical images were taken at 5X resolution using a Nikon LV150M microscope and stitched together in photoshop.

finite standard deviation (σ_G) on the signal transduced by T_{PT} to generate noisy voltage, V_{PSV} , the MoS₂ FETs used as the spiking neurons (T_{SN}) encrypt the noisy presynaptic information i.e., V_{PSV} into post-synaptic current spikes (I_{PSC}) using reconfigurable encoding threshold (V_{ET}), and finally, the two remaining MoS₂ FETs, i.e., photo selector transistor (T_{PST}) and encoding selector transistor (T_{EST}) operate as the individual selector switches for the photosensing and the encoding operations,

Figure R3. Optical images of a) fully integrated crossbar array of crypto engines with each crypto engine comprising of five MoS₂ FETs (5T cell) to accomplish sensing, storage, and encoding functionalities, and b) individual MoS₂ FETs, which are locally back-gated using a stack comprising of 50 nm Al₂O₃ on 40/30 nm Pt/TiN. All back-gate islands were placed on SiO₂/p⁺⁺-Si substrate. Equivalent circuit diagram for d) the crossbar array with column and row select lines and e) each crypto engine. MoS₂ FET used as T_{PT} mimics sensory neurons and transduce optical information into persistent photoconductance (G_{PT}), MoS₂ FET used as T_{WGNA} emulates noisy synapses, MoS₂ FET used as T_{SN} imitates spiking neurons converting noisy presynaptic information i.e., V_{PSV} into post-synaptic current spikes (I_{PSC}), and the remaining 2 MoS₂ FETs, T_{PST} and T_{EST} , operate as the individual selector switches for the photosensing and the encoding operations, respectively.

respectively. In Fig. R3d, T_{PST} and T_{PT} are connected in series at the node, N_3 , T_{PT} and T_{WGNA} are connected in series at the node, N_5 , which is also connected to the local back-gate of T_{SN} , and finally, T_{EST} and T_{SN} are connected in series at the node, N_{10} . The nodes, N_2 , N_4 , N_6 , and N_9 , respectively, serve as the local back-gate terminals of T_{PST} , T_{PT} , T_{WGNA} , and T_{EST} , node N_1 serves as the drain terminal of T_{PST} , node N_8 serves as the drain terminal of T_{EST} , and node N_7 serves as the common source terminal for T_{WGNA} and T_{SN} . As shown in Fig. R3c, nodes N_2 , N_4 , N_6 , and N_9 from the crypto engines in a given column are connected to common $V_{\text{N}2}$, $V_{\text{N}4}$, $V_{\text{N}6}$ and $V_{\text{N}9}$ lines, respectively, and nodes N_1 , N_7 , and N_8 from the crypto engines in a given row are connected to common $V_{\text{N}1}$, $V_{\text{N}7}$, and $V_{\text{N}8}$ lines, respectively. This allows us to select any crypto engine corresponding to a given row and column. Note that the overlapping connections between the horizontal and vertical metal lines are separated lithographically by depositing an insulating layer of alumina (Al_2O_3) at the cross points as described in the *Methods* section.

2. The authors need to evaluate the yield of the array fabrication, especially the transfer step. They should explain how the wrinkles and cracks affect the device yield since the wrinkles are obvious in Fig. 2b.

Reviewer has raised a valid concern. It is important to mention the yield of the fabricated devices. Since we use PMMA assisted wet transfer technique, during the transfer of the film from the growth substrate to the desired substrate, there could be a possibility of strain, wrinkles, cracks, and damage to the film. These effects could negatively impact the device performance metrics

Figure R4. Transfer characteristics of the 320 monolayer MoS_2 FETs used in the crypto engine.

such as mobility, ON current, subthreshold slope, threshold voltage, and sometimes failure of devices. Over the last one year, we have thrived to improve the transfer process and we are now able to largely avoid wrinkles or cracks during the transfer of the films through extra care. We can now achieve near ideal yield close to 100% during the fabrication process. Fig.R4 shows the transfer characteristics of the fabricated 320 devices. Although there exist device-to-device variations, efforts have been put forward to minimize the variation and improve the efficacy of the fabricated devices.

3. It seems that the authors only use one photodetector transistor to measure the IPH with a 64*1 array of LED voltages. However, for real applications, they should use 64 different devices to measure the IPH, because the device variation may also affect the pattern 'N' and the encoded image.

Reviewer has raised an excellent point. We have now fabricated 64 photodetectors and discussed the impact of device-to-device variation in the photoresponse on the crypto engine in the revised manuscript as detailed below.

Fig. R5a shows the pre-and post-illumination transfer characteristics of 64 monolayer MoS₂ FETs used as T_{PT} corresponding to each of the 8×8 arrays of the crypto engine. Fig. R5b compiles the device-to-device variation in photogating effect and Fig. R5c shows the corresponding 8×8 colormaps of distribution of the ratio of post-illumination photoconductance to dark conductance (r_{PH}) measured at $V_{BG} = 0$ V. The mean and standard deviation values were found to be $\sim 1.6 \times 10^4$ and $\sim 3.1 \times 10^4$, respectively. Fig. R5d shows the colormap of persistent photoconductivity (G_{PT}) measured at $V_{BG} = 0$ V, when the T_{PT} corresponding to the 8×8 arrays of the crypto engines are exposed to 8×8 pixelated images of the letters, 'L', 'M', 'N', and 'P', obtained through the LED illumination for $t_{illumination} = 100$ ms with $V_{illumination} = -2$ V. The bright pixels correspond to $I_{LED} = 20$ mA. Clearly, MoS₂ FETs integrated with the crypto engines can accurately transcribe the optical information into an electrical response. Note that the device-to-device variation in photogating effect is naturally present in our experimental demonstration. Yet, the encoding and decoding process appears to function as expected.

Figure R5. Transfer characteristics of 64 monolayer MoS₂ FETs across the 8 × 8 array of the crypto engines measured in dark (pink) and post illumination (yellow). b) Compiled device-to-device variation in photoresponse. c) Colormap of distribution of ratio of post-illumination photoconductance to dark conductance (r_{PH}) measured at $V_{BG} = 0$ V. The mean and standard deviation values for MR were found to be 1.6×10^4 and standard deviations of 3.6×10^4 , respectively. d) Colormap of persistent photoconductivity (G_{PT}) measured at $V_{BG} = 0$ V, when the MoS₂ FETs used as sensory neurons (T_{PT}) corresponding to the 8 × 8 array of the crypto engines are exposed to 8 × 8 pixelated images of the letters, ‘L’, ‘M’, ‘N’, and ‘P’, obtained through the LED illumination for $t_{illum} = 100$ ms with $V_{illum} = -2$ V. The bright pixels correspond to $I_{LED} = 20$ mA.

4. In Fig. 3a, it seems that the CA or WGNA is a physical adder that relies on Kirchhoff's law, indicating that at least three probes should be connected to the adder to connect two input terminals (IPH and Inoise) and one output terminals (Iout or VPSV) simultaneously. However, in Fig. S1, only two probes are connected to the crypto engine which is not consistent.

We apologize for the confusion. We have redesigned our experimental setup and have now fabricated a fully integrated 8×8 crossbar array of the crypto engines to encode 8×8 -pixel images with each crypto engine integrating five locally gated transistors (5T cell) as detailed in your response to your earlier comments as shown in Fig R1-3. Our entire hardware platform utilizes a total of 320 MoS₂ FETs. The revised manuscript provides a detailed description of how the different building blocks (photodetectors, memory devices, and other transistors) are electrically connected (Fig. R3) and how each element and the entire crypto engines operate. **Supplementary video 1** shows the video demonstration of electrical probing to our crypto engine using automated probe station.

5. In Fig. S1, the LED can shine lights on the detector but also the encoder. The authors need to explain how the light from LED affects the noise current in the WGNA.

Reviewer has raised an excellent point. It is true that when the detector is exposed to light, the encoder could also observe a change in the device characteristics. However, in the revised circuit, we have exploited photogating effect to sense the input information. Fig. R6 shows the transfer characteristics for a representative monolayer MoS₂ FET before and after illumination from a blue LED with input currents ranging from $I_{LED} = 0.02$ mA (low-brightness) to $I_{LED} = 20$ mA (high-brightness) at different $V_{BG} = V_{illum}$ for $t_{illum} = 1$ second. Two distinctive types of photoresponse are seen in MoS₂ FETs. For $V_{illum} > 0$ V, i.e. illuminations in the on-state and the subthreshold regime, the ratio of post-illumination conductance to dark conductance (r_{PH}), $r_{PH} = 1$, whereas, for $V_{illum} < 0$ V, i.e. illuminations in the off-state, $r_{PH} \gg 1$. This can be explained by the fact that during illumination in the on-state the photocarrier generated in the MoS₂ channel are swept across by the applied V_{DS} and hence there is no persistent photocurrent beyond the optical exposure. However, for $V_{illum} < 0$ V, persistent photoconductivity emerges as photocarriers get

Figure R6. Transfer characteristics for a representative monolayer MoS₂ FET before and after illumination from a blue LED with input currents ranging from $I_{LED} = 0.02$ mA (low-brightness) to $I_{LED} = 20$ mA (high-brightness) at different $V_{BG} = V_{illum}$ for $t_{illum} = 100$ ms.

trapped at and near the MoS₂/dielectric interface leading to the shift in the post-illumination device characteristics. The detrapping process can take several hours. Higher I_{LED} , more negative V_{illum} , and longer t_{illum} naturally result in more photocarrier trapping and hence larger shifts in the device characteristics leading to higher values of r_{PH} . [1] Since the T_{SN} and T_{WGNA} are operated in the subthreshold regime, they exhibit little-to-no variation in electrical characteristics when exposed to light, while the T_{PT} is operated in the deep-off state ($V_{illum} = -4$ V), persistent photoconductivity emerges and can sense the input information.

6. A key piece of information is missing that is how the authors store the measured IPH and Inoise (64*1 arrays). If they sum the I-PH and Inoise directly after measurement, please go back to Q4 and 5. If they first measure the current and store them and use the measured current as inputs to the CA to get encoded output current, they actually use external memories to store the temporal information for next-step signal processing, which is not a fully MoS₂-based crypto engine. In this case, they need to clearly mention this data storage step.

Reviewer has raised a valid concern. The reviewer was right about the fact we need some extra peripheral circuits to temporarily store the obtained information. Therefore we have redesigned our crypto engine as detailed in your response to your earlier comments as shown in Fig R1-3. In our revised design of 8×8 array of the crypto engines, each crypto engine monolithically integrates five locally gated transistors (5T cell) to accomplish sensing and encoding functionalities. Our entire hardware platform utilizes a total of 320 individually gated MoS₂ FETs. The revised manuscript provides a detailed description of how the different building blocks (photodetectors, memory devices, and other transistors) are electrically connected (Fig. R3) and how each element and the entire crypto engines operate. This circuit eliminate the need for the

extra peripheral circuits to store the information thereby reasserting our claims of using monolayer MoS₂.

7. The explanation about the floating gate (FG) is not convincing. (i) If the authors use heavily doped p⁺⁺ silicon as the common back gate, the contact between TiN and Si should be ohmic contact since the high doping concentration already makes tunneling happen. (ii) Even if we assume the existence of the Schottky barrier (SB) at the interface, the forward direction for holes in this SB is from Si to TiN, which means the trapped negative charges can go through the SB easily (from TiN to Si). In this case, the SB cannot store the trapped negative charges in the TiN layer.

We would like to thank the reviewer for raising a valid concern, which lead us to investigate deeper into the origin of non-volatile memory retention in our back-gate stack comprising of 50 nm atomic layer deposition grown (ALD) Al₂O₃ on Pt/TiN/p⁺⁺-Si. We agree with the reviewer that since p⁺⁺ Si, TiN and Pt are all conductive and flow of charges between TiN and Si should be relatively easy. Our initial hypothesis was based on the assumption that since TiN is known to form non-conductive native oxide on the surface,

Figure R7. Current versus voltage characteristics across the Pt/TiN/p⁺⁺-Si stack demonstrate Ohmic nature of carrier transport across the stack. Inset shows the schematic of the Pt/TiN/p⁺⁺-Si stack.

there exists a tunneling barrier between TiN and Pt, which ensures charge trapping and retention inside the Pt layer. However, direct measurement of current *versus* voltage characteristics across the Pt/TiN/p⁺⁺-Si stack revealed Ohmic nature of carrier transport across the stack as seen in Fig. R7. Therefore, we must rule out charge storage in the Pt electrode layer. In fact, when we fabricated

individually gated MoS₂ FETs with Al₂O₃ on Pt/TiN/SiO₂/p⁺⁺ Si as the back-gate stack and applied the back-gate voltage to the Pt electrode we were still able to program the devices in various analog conductance states with non-volatile memory retention, reinforcing the fact that charge storage is not mediated by the Pt electrode and hence must be related to carrier trapping and detrapping in our gate oxide i.e. in the Al₂O₃ layer.

8. The x-axis (time) of Fig. 2f and Fig. 2h are missing, which are important for retention results. The authors need to show the time interval between the measurements after the VP pulse.

Reviewer has raised a valid concern. The memory states are sampled every 1 second for a total duration of 10 s after each V_p pulse. In the revised manuscript we have also performed some more extensive memory characterization such as retention time and endurance of our memory gate-stack as discussed below.

Fig. R8a-b, respectively, show the post-programmed and post-erased transfer characteristics of a representative MoS₂ FET when subjected to negative “Write” (V_p) and positive “Erase” (V_E) voltage pulses of different amplitudes applied to the local back-gate electrode, each for a duration of $\tau_{p/E} = 1$ second. The shift in the transfer characteristics can be attributed to charge trapping/detrapping at and near the MoS₂/Al₂O₃ interface. Negative shift in the in the transfer characteristics with increasing magnitude of V_p and positive shift with increasing magnitude of V_E are indicative of electron trapping and de-trapping in the local back-gate stack, respectively. Interestingly, the trapping and de-trapping processes were found to be non-volatile as evident from the retention measurements displayed in Fig. R8c-d for 5 representative post-programmed (G_p) and post-erased (G_E) conductance states, respectively, for 100 seconds. We also examined long-term memory retention for two representative analog conductance states for $\sim 10^4$ seconds as shown in Fig R8e-f. The memory ratio (MR) between these two states was found to change from $\sim 6 \times 10^2$ to $\sim 2 \times 10^2$ following an exponential decay with a time constant of 7.6×10^3 seconds. The projected time before the MR reaches 1, i.e. these two states become indistinguishable is found to be ~ 14 hours. The MR can be reinstated by reprogramming the devices every few hours or as necessary. This will necessitate some peripheral timing circuits. Improving the memory retention through the

optimization of our back-gate stack can eliminate the need for extra peripherals. Similarly, Fig. R8g-h shows the memory endurance for 2×10^3 cycles for two representative analog conductance

Figure R8. a) Post-programmed and b) post-erased transfer characteristics of a representative MoS_2 FET when subjected to negative “Write” (V_P) and positive “Erase” (V_E) voltage pulses of different amplitudes applied to the local back-gate electrode, each for a duration of 100 ms, respectively. Non-volatile retention of 5 representative c) post-programmed (G_P) and d) post-erased (G_E) conductance states for 100 seconds. e) Long-term retention of two representative post-programmed analog conductance states for $\sim 10^4$ seconds. f) Evolution of the memory ratio (MR) between these two programmed conductance states (brown) and the corresponding exponential fitting (blue) given by, $MR = MR_0 \exp(-t/\tau)$, where $MR_0 = 600$, $\tau = 7.6 \times 10^3$ s. The projected time before the MR reaches 1, i.e., these two states become indistinguishable is found to be ~ 14 hours. g) Memory endurance when successively programmed and erased to achieve two representative analog conductance states for 2×10^3 cycles with programming and erase voltage of $V_P = -9$ V and $V_E = 10$ V respectively applied for duration of 100 ms. h) Evolution of memory ratio (MR) as a function of memory cycle (green) and the corresponding power law fitting (pink) given by, $MR = MR_0 t^\gamma$, with $\gamma \sim -0.1$. The projected memory endurance before the MR reaches 1 is found to be 5×10^8 cycles.

states. MR was found to change from ~ 7.5 to ~ 5 following a power law decay with an exponent of ~ -0.1 . The projected memory endurance before the MR reaches 1 is found to be 5×10^8 cycles, which is better than the state-of-the-art FLASH memory devices.

9. Fig. S4 is not convincing enough. (i) The authors need to mention the V_p pulse used here and the duration. (ii) The difference between Fig. S5a and S5b is not as obvious as that shown in the main text. Here, the V_{th} shift is less than 0.5 V. However, in Fig. 2f, the V_{th} shift can reach 2 V. The author should use the extreme condition (the largest V_{th} shift to the programmed device) to prove the negligible effect to unselected devices. (iii) The authors need to briefly explain the reason why there is no crosstalk effect to unselected devices.

Reviewer concern is noted. However, in the revised manuscript, we have fabricated locally back-gated islands instead of global back-gate as shown in Fig.R3. Since each device is individually gated there will be no crosstalk among the devices and the effect of programming is local.

10. The brute force trials (BFT) calculated here are not appropriate to evaluate the efficiency of the crypto engine. The information is an 8×8 'N' image, where pixels have interconnections and hidden information. To decode the information of this image, we do not need to exactly get all the pixels information correctly. The author should not emphasize the 'astronomical value'. The simulation of artificial neural networks (ANN) here is more reliable.

We agree with the reviewer. To demonstrate the strength of the encryption process, a more complex machine learning algorithm is implemented in a much deeper neural network model. In the revised manuscript, we have tested the encryption strength using deep neural network (DNN) as shown in Fig. R9. DNNs have shown remarkable success in various applications ranging from image recognition, pattern classification to defeating professional players in the game of "Go"[2]. DNNs consists of one input, one output, and many hidden layers (N) and each layer in a network contains a certain number of neurons. The neurons in a layer and its consecutive layer are connected by synapses and the strength of the synapses, i.e., synaptic weights are trained using back-propagation algorithms. The deeper the network is, i.e., for higher N , the more complex dataset can be trained with better accuracy. Fig. R9a shows the schematic of a DNN trained to recognize MNIST data set. The input layer consists of 784 neurons where the normalized pixel values of the images (28×28 pixels) in the MNIST dataset are sent and the output layer consists of 10 neurons that correspond to digits from 0 to 9. Output neurons perform the SoftMax activation function, and the winner is determined based on the maximum value. We have used $N_{\text{Layer}} = 1, 2, 3, 4$ hidden layers with 300, 200, 100, and 50 neurons in the corresponding layers, respectively. Gradient decent algorithm is used to train the DNN using 60,000 images with a learning rate of 0.003 and rectified linear unit (ReLU) as the activation function and remaining 10,000 images are used for testing the inference accuracy. Fig. R9b shows the training and inference accuracy as a function of N_{Layer} and number of epochs. Training accuracy of 100% and testing accuracy of $> 98\%$ was achieved beyond 50 epochs irrespective of N_{Layer} . Note that higher N_{Layer} achieves better training and inference accuracies with lesser number of epochs. Next, we added white

Gaussian noise (WGN) to 10,000 MNIST testing images and binarized them at a threshold of 1.5 mimicking the encryption process of our MoS₂ FET-based crypto engines. Fig. R9c shows some

Figure R9. a) Schematic of a DNN trained to recognize MNIST data set. The input layer consists of 784 neurons where the normalized pixel values of the images (28×28 pixels) in the MNIST dataset are sent and the output layer consists of 10 neurons that correspond to digits from 0 to 9. Output neurons perform the SoftMax activation function, and the winner is determined based on the maximum value. We have used $N_{Layer} = 1, 2, 3, 4$ hidden layers with 300, 200, 100, and 50 neurons in the corresponding layers, respectively. Gradient decent algorithm is used to train the DNN using 60,000 images with a learning rate of 0.003 and rectified linear unit (ReLU) as the activation function and remaining 10,000 images are used for testing the inference accuracy. b) The training and inference accuracy as a function of the number of hidden layers (N_{Layer}) and number of epochs for a deep neural network for MNIST digit classification. Training accuracy of 100% and testing accuracy of $> 98\%$ was achieved beyond 50 epochs irrespective of N_{Layer} . c) Representative MNIST images superimposed with white Gaussian noise (WGN) of different standard deviation (σ) and subsequently binarized at a threshold of 1.5 mimicking the MoS₂ FET-based encryption process. d) Average inference accuracy for 10,000 encrypted images as a function of σ and number of hidden layers (N_{Layer}) of a deep neural network (DNN), which was trained to recognize MNIST data set for digit classification using 60,000 clear images. Irrespective of σ , the inference accuracy remains low indicating the robustness of our bio-inspired encryption to trained DNNs.

example of encoded MNIST images for different standard deviation (σ) of the WGN. Fig. R9d shows the inference accuracy for the encrypted images as a function of σ for different N_{Layer} . Interestingly, the accuracy values are found to be significantly low irrespective of σ and N_{Layer} , indicating the robustness of our proposed encryption scheme to trained DNNs. Also note that we have removed the word astronomical in the revised manuscript.

Minor comments:

During measurement, the authors need to clearly mention how they measure the current for the 64 noise transistors. They choose to either (i) program all the devices first and measure them or (ii) program one device and measure immediately and move to the second one.

Reviewer has raised a valid concern. At first, we program all the T_{WGNA} 's to random conductance states following Gaussian distribution and later measure the encrypted information from the T_{SN} .

Unit is missing (1.5 V) in the last sentence in Page 13.

We apologize for the typographical error. We have fixed it in the revised manuscript.

Reviewer #2 (Remarks to the Author):

The main expertise of the authors is in the development of MoS₂ FETs. In this paper, they experiment and develop an IOT edge sensor. The authors claim that this new design provides encryption. The main problem with the paper is the claimed security level. The first 6 references are related to security, the rest is related to MoS₂ FET devices and designs. These first 6 references are absolutely not state of the art. RSA is never used in sensor nodes. Much more low power and low energy solutions exist, for instance based on elliptic curves. Public key crypto systems that fit in the power budget of passive RFIDs have been developed.

We would like to thank the reviewer for his valuable insights. We apologize for missing some the recent references on IoT security. We have now performed an extensive literature survey and included the relevant references in the revised manuscript [3-9]. We agree with the reviewer that there exist low power security solutions such as elliptic curve cryptography (ECC) [3-9]. However, the reviewer will agree that the implementation of ECC algorithms even when they are modified to reduce the circuit/architecture complexity require significant hardware resources for both compute and memory. Furthermore, these systems sense the information from unprotected edge devices, making them vulnerable to possible tampering of the information even before the ECC based encryption [10, 11]. In contrast, our encryption scheme offers in-sensor security with very limited hardware resources, namely, the 5T cell module as demonstrated in the revised manuscript. This is owing to the unique property of 2D material-based devices, which allow them to be used as photosensors, encoders, as well as storage devices. Furthermore, in spite of the encryption algorithm being simple, we show that the encrypted information is resilient to advanced machine learning attacks based on deep neural networks (DNNs). Nevertheless, our objective is not to undermine the existing security solutions for IoT sensor nodes. Instead, we offer another low-power alternative security solution for IoT edge devices, which can be integrated with the sensor itself.

The proposed solution only provides encryption, i.e. hiding of the information. The solution still cannot address what the authors call "loss, misuse and manipulation" in the introduction. It seems that the attacker is only allowed 'passive eavesdropping' meaning that he can only observe the images that pass by.

We agree with the reviewer that our proposed security scheme offer only information encryption. We have removed the phrase “loss, misuse and manipulation” from the revised manuscript. The reviewer is also correct that the attacker is only allowed 'passive eavesdropping'. Note that investigation of various attack models is beyond the scope of the current manuscript and will be investigated in our future work. We have mentioned this in the revised manuscript.

Figure 4, subfigure (d), shows the population means and the detectivity. This detectivity reaches its maximum of around 45% for a σ_V around 0.75. At this σ_V , subfigure (e) shows that the attacker needs around 10^{20} Brute Force Trials. This corresponds to 66 bits of security. Modern IOT devices require 128 bits of security for brute force attacks.

Reviewer has raised a valid concern regarding the number of brute force trials and number of bits. We agree with the reviewer that the BFTs of $\sim 10^{20}$ is decoded in recent years. However in our revised full scale demonstration, the maximum detectivity value was observed to be ~ 0.29 at a standard deviation (σ_G) of 0.4, which corresponds to a BFTs of $\sim 3.3 \times 10^{34}$, which is relatively high and correspond to 115 bits of security. Furthermore, the number of BFTs increases exponentially with S (bit length) as shown in Fig. R10. To further elucidate the efficacy of the encrypted information we have implemented a more complex machine learning algorithm in a much deeper neural network model as shown in Fig. R11. DNNs have shown remarkable success in various applications ranging from image recognition, pattern classification to defeating professional players in the game of “Go”[2]. DNNs consists of one input, one output, and many

Figure R10. Number of brute force trials (BFTs) required to decode the letter 'N' for different image size (S) as a function of σ_G .

hidden layers (N) and each layer in a network contains a certain number of neurons. The neurons in a layer and its consecutive layer are connected by synapses and the strength of the synapses, i.e., synaptic weights are trained using back-propagation algorithms. The deeper the network is, i.e., for higher N , the more complex dataset can be trained with better accuracy. Fig. R11a shows the schematic of a DNN trained to recognize MNIST data set. The input layer consists of 784 neurons where the normalized pixel values of the images (28×28 pixels) in the MNIST dataset are sent and the output layer consists of 10 neurons that correspond to digits from 0 to 9. Output neurons perform the SoftMax activation function, and the winner is determined based on the maximum value. We have used $N_{\text{Layer}} = 1, 2, 3, 4$ hidden layers with 300, 200, 100, and 50 neurons

Figure R11. a) Schematic of a DNN trained to recognize MNIST data set. The input layer consists of 784 neurons where the normalized pixel values of the images (28×28 pixels) in the MNIST dataset are sent and the output layer consists of 10 neurons that correspond to digits from 0 to 9. Output neurons perform the SoftMax activation function, and the winner is determined based on the maximum value. We have used $N_{\text{Layer}} = 1, 2, 3, 4$ hidden layers with 300, 200, 100, and 50 neurons in the corresponding layers, respectively. Gradient decent algorithm is used to train the DNN using 60,000 images with a learning rate of 0.003 and rectified linear unit (ReLU) as the activation function and remaining 10,000 images are used for testing the inference accuracy. b) The training and inference accuracy as a function of the number of hidden layers (N_{Layer}) and number of epochs for a deep neural network for MNIST digit classification. Training accuracy of 100% and testing accuracy of $> 98\%$ was achieved beyond 50 epochs irrespective of N_{Layer} . c) Representative MNIST images superimposed with white Gaussian noise (WGN) of different standard deviation (σ) and subsequently binarized at a threshold of 1.5 mimicking the MoS₂ FET-based encryption process. d) Average inference accuracy for 10,000 encrypted images as a function of σ and number of hidden layers (N_{Layer}) of a deep neural network (DNN), which was trained to recognize MNIST data set for digit classification using 60,000 clear images. Irrespective of σ , the inference accuracy remains low indicating the robustness of our bio-inspired encryption to trained DNNs.

in the corresponding layers, respectively. Gradient decent algorithm is used to train the DNN using 60,000 images with a learning rate of 0.003 and rectified linear unit (ReLU) as the activation function and remaining 10,000 images are used for testing the inference accuracy. Fig. R11b shows the training and inference accuracy as a function of N_{Layer} and number of epochs. Training accuracy of 100% and testing accuracy of $> 98\%$ was achieved beyond 50 epochs irrespective of N_{Layer} . Note that higher N_{Layer} achieves better training and inference accuracies with lesser number of epochs. Next, we added white Gaussian noise (WGN) to 10,000 MNIST testing images and binarized them at a threshold of 1.5 mimicking the encryption process of our MoS₂ FET-based crypto engines. Fig. R11c shows some examples of encoded MNIST images for different standard deviation (σ) of the WGN. Fig. R11d shows the inference accuracy for the encrypted images as a function of σ for different N_{Layer} . Interestingly, the accuracy values are found to be significantly low irrespective of σ and N_{Layer} , indicating the robustness of our proposed encryption scheme to trained DNNs. Therefore, the encryption can be considered to be secure from an eavesdropper with finite resources.

Reviewer #3 (Remarks to the Author):

The authors demonstrate an all-in-one biomimetic IoT hardware platform based on multifunctional MoS₂ FETs that integrates sensing, non-volatile storage, and security. The sensing elements are realized in a MoS₂ field effect transistor-technology with a programmable gate stack that can be used as a photodetector (PD). The term biomimetic stems from the analogy of the cryptoengine to an artificial neural encoder. In principle the mechanism of sensing, encoding and decoding is explained well, but some information especially on transistor and circuit level is missing. For example, both the WGNA and the AN are realized with thin film MoS₂ transistors (in form of a current adder and look up table as explained later), but what is the exact schematic? Also, is there a difference in terms of layout, physical realization, or are all MoS₂ transistors identical? How is the integration done between WGNA and AN? Are they connected to independent voltage sources?

We would like to thank the reviewer for appreciating our explanations. We are sorry for missing the circuit level information. We have revised our manuscript extensively through the realization

The all-in-one bio-inspired crypto chip

Figure R12. Optical image of the all-in-one bio-inspired crypto chip.

of 8×8 array of the crypto engines with each crypto engine monolithically integrating five locally gated transistors (5T cell) to accomplish sensing and encoding functionalities. Our entire hardware platform utilizes a total of 320 MoS₂ FETs.

Fig. R12 and Fig. R13, respectively, show the optical images of the entire chip and the 8×8 array of the crypto engines. Fig. R14a-b, respectively, show the optical images of each crypto engine

Figure R13. Optical image of the crossbar array of the crypto engines. Note that several (six) optical images were taken at 5X resolution using a Nikon LV150M microscope and stitched together in photoshop.

with five MoS₂ FETs (5T cell) that integrates sensing, storage, and encoding functionalities, and of individual MoS₂ FETs, which are locally back-gated using a stack comprising of atomic layer deposition (ALD) grown 50 nm Al₂O₃ on sputter deposited 50/20 nm Pt/TiN. All back-gate islands were placed on a commercially purchased SiO₂/p⁺⁺-Si substrate. The Al₂O₃/Pt/TiN gate islands not only allow non-volatile programming of our MoS₂ FETs but also enhances the photoresponse of MoS₂ FETs owing to the phenomenon of gate-tunable persistent photoconductivity when subjected to right polarity and magnitude of local back-gate biases. This, in turn, empowers our hardware platform to enable in-memory computing and near-sensor security, which are presently

Figure R14. Optical images of a) fully integrated crossbar array of crypto engines with each crypto engine comprising of five MoS₂ FETs (5T cell) to accomplish sensing, storage, and encoding functionalities, and b) individual MoS₂ FETs, which are locally back-gated using a stack comprising of 50 nm Al₂O₃ on 40/30 nm Pt/TiN. All back-gate islands were placed on SiO₂/p⁺⁺-Si substrate. Equivalent circuit diagram for d) the crossbar array with column and row select lines and e) each crypto engine. MoS₂ FET used as T_{PT} mimics sensory neurons and transduce optical information into persistent photoconductance (G_{PT}), MoS₂ FET used as T_{WGNA} emulates noisy synapses, MoS₂ FET used as T_{SN} imitates spiking neurons converting noisy presynaptic information i.e., V_{PSV} into post-synaptic current spikes (I_{PSC}), and the remaining 2 MoS₂ FETs, T_{PST} and T_{EST}, operate as the individual selector switches for the photosensing and the encoding operations, respectively.

lacking for the conventional silicon technology as well as emerging solutions based on memristive crossbar architectures. Fig. R14c-d, respectively, show the circuit schematic for the crossbar architecture and each crypto engine. In short, MoS₂ FETs used as the photo transistor (T_{PT}) mimic sensory neurons and transduce optical information into persistent photoconductance (G_{PT}), MoS₂ FETs used as the white Gaussian noise adder (T_{WGNA}) emulate noisy synapses and are pre-programmed into random conductance states (G_{WGNA}) to superimpose white Gaussian noise of finite standard deviation (σ_G) on the signal transduced by T_{PT} to generate noisy voltage, V_{PSV} , the MoS₂ FETs used as the spiking neurons (T_{SN}) encrypt the noisy presynaptic information i.e., V_{PSV} into post-synaptic current spikes (I_{PSC}) using reconfigurable encoding threshold (V_{ET}), and finally, the two remaining MoS₂ FETs, i.e., photo selector transistor (T_{PST}) and encoding selector transistor (T_{EST}) operate as the individual selector switches for the photosensing and the encoding operations, respectively. In Fig. R14d, T_{PST} and T_{PT} are connected in series at the node, N_3 , T_{PT} and T_{WGNA} are connected in series at the node, N_5 , which is also connected to the local back-gate of T_{SN} , and finally, T_{EST} and T_{SN} are connected in series at the node, N_{10} . The nodes, N_2 , N_4 , N_6 , and N_9 , respectively, serve as the local back-gate terminals of T_{PST} , T_{PT} , T_{WGNA} , and T_{EST} , node N_1 serves as the drain terminal of T_{PST} , node N_8 serves as the drain terminal of T_{EST} , and node N_7 serves as the common source terminal for T_{WGNA} and T_{SN} . As shown in Fig. R14c, nodes N_2 , N_4 , N_6 , and N_9 from the crypto engines in a given column are connected to common V_{N2} , V_{N4} , V_{N6} and V_{N9} lines, respectively, and nodes N_1 , N_7 , and N_8 from the crypto engines in a given row are connected to common V_{N1} , V_{N7} , and V_{N8} lines, respectively. This allows us to select any crypto engine corresponding to a given row and column. Note that the overlapping connections between the horizontal and vertical metal lines are separated lithographically by depositing an insulating layer of alumina (Al_2O_3) at the cross points as described in the *Methods* section.

Also, is the generation of white noise really guaranteed for MoS₂? Many thin film transistors based on organic and inorganic channel materials show $1/f$ or even $1/f^\gamma$ noise for low frequencies. What is the typical delay of the used MoS₂ transistors (e.g. obtained from pulsed measurements or by creating inverter cells)?

We believe there is some misunderstanding. We are not exploiting the intrinsic noise from the MoS₂ FETs. The white Gaussian noise from the MoS₂ FETs was obtained by programming the

T_{WGNA} to random conductance states following a Gaussian distribution with different standard deviations (σ_G). So that measuring current at a particular gate voltage from WGNAs follows a Gaussian distribution. The delay in our experimental setup relates to the exposure time of the T_{PT} to light. The T_{PT} is exposed to light for a t_{illum} of 100 ms.

I believe the significance of this work is solid, however there are many works on optical artificial neural networks and neuromorphic computing architectures, that could be competitors for the artificial encoding presented here with possibly higher information complexity. Just a few references, which are by far not comprehensive:

Ref. 1: Feldmann, J., Youngblood, N., Wright, C.D. et al. All-optical spiking neurosynaptic networks with self-learning capabilities.

(Nature 569, 208–214 2019). <https://doi.org/10.1038/s41586-019-1157-8>

Ref. 2: Lei Yin, Cheng Han, Qingtian Zhang, Zhenyi Ni, Shuangyi Zhao, Kun Wang, Dongsheng Li, Mingsheng Xu, Huaqiang Wu, Xiaodong Pi, Deren Yang, Synaptic silicon-nanocrystal phototransistors for neuromorphic computing,

Nano Energy 63 (2019), doi.org/10.1016/j.nanoen.2019.103859.

We would like to thank the reviewer for pointing out the references and considering our work to be solid. We agree with the reviewer that there have been many works related to optical artificial neural networks and neuromorphic computing architectures. However, to the best of our knowledge, there has been no demonstration that has monolithically integrated compute, sensing, storage, and security. A benchmarking table shows the integration of different functionalities achieved in recent demonstrations highlighting the significance and novelty of our work. Furthermore, in our revised manuscript, we have now demonstrated circuit-level integration and array level realization of the proposed crypto engine.

Table R1: Benchmarking Emerging IoT Platforms

Table R1: Benchmarking Emerging IoT Platforms					
	Material	Sensing	Storage	Security	Reference
2D	MoS ₂	Y	N	N	[12]
	MoS ₂ , WS ₂ , MoSe ₂ , WSe ₂	N	Y (Electronic)	N	[13]
	WSe ₂ /MoS ₂ /h-BN/HfS ₂ /WSe ₂ /MoS ₂	N	Y (Electronic)	N	[14]
	Graphene/MoS _{2-x} O _x /Graphene	N	Y (Electronic)	N	[15]
	Graphene/MoS ₂	N	Y (Electronic)	N	[16]
	Graphene/h-BN/MoS ₂	N	Y (Electronic)	N	[17]
	Graphene/h-BN/MoS ₂	N	Y (Electronic)	N	[18]
	Graphene/MoS ₂	Y	Y (Optical)	N	[19]
	h-BN/WSe ₂ - h-BN/WSe ₂	Y	Y (Optical)	N	[20]
	MoS ₂ /PTCDA	Y	Y (Optical)	N	[21]
	MoS ₂ /Au-Nano Particles	Y	Y (Optical)	N	[22]
	WSe ₂ /h-BN	Y	Y (Optical)	N	[23]
	MoS ₂ /PbS	Y	Y (Optical)	N	[24]
	BP/Al ₂ O ₃	N	Y (Electronic)	N	[25]
	BP/h-BN/MoS ₂	N	Y (Electronic)	N	[26]
	BP/Al ₂ O ₃ /BP/Al ₂ O ₃	N	Y (Electronic)	N	[27]
	MoS ₂ /Metal Nano Crystal	N	Y (Electronic)	N	[28]
	MoS ₂	Y	N	N	[29]
	MoS ₂	Y	Y (Electronic)	N	[30]
	MoS ₂ /PZT	Y	Y (Electronic)	N	[31]
WSe ₂	Y	N	N	[32]	
MoO _x /MoS ₂	N	Y (Electronic)	N	[33]	
MoS ₂	N	N	Y	[34]	
Oxide Based Memristors	Ag:SiO ₂ or MgO/HfO ₂ :Ag	N	Y (Electronic)	N	[35]
	TiN/TaO _x /HfAl _y O _x /TiN	N	Y (Electronic)	N	[36]
	ITO/LaAlO ₃ /SrTiO ₃	N	Y (Electronic)	N	[37]
	Ag ₂ S	N	Y (Electronic)	N	[38]
	Indium Gallium Zinc Oxide (IGZO)	N	Y (Electronic)	Y	[39]
	Al ₂ O ₃ /TiO _{2-x}	N	Y (Electronic)	Y	[40]
	Ag:SiO ₂	N	Y (Electronic)	Y	[41]
	Ag:SiO ₂	N	Y (Electronic)	Y	[42]
	TiO ₂	N	Y (Electronic)	Y	[43]
Phase Change Materials	GeSbTe(GST)	Y	N	N	[44]
Nano Crystals (NC)	B-doped Si NC		Y (Optical)	N	[45]
2D	This Work	Y	Y (Electronic)	Y	

The quality of data and especially the statistical analysis appears very reasonable and sound.

We are glad that the reviewer finds our data and the statistical analysis to be reasonable and sound. We are humbled by the kind words from the reviewer.

It would be nice to have some evidence for the robustness of the chosen design, meaning figure of merits such as reliability w.r.t. to typical environmental noise, temperature etc. However, these values need extensive characterization data and if not available, could maybe be simply commented based on the authors experience with the technology in the text.

Reviewer has raised an excellent point. We have commented on the impact of environmental noise, temperature etc. on the encryption process in the revised manuscript. We have also included the device-to-device variation in the MoS₂ FET transfer characteristics, programmability, and photoresponse in the Supplementary Information. We have found that the device-to-device variation is dominant compared to environmental noise. Interestingly, the device-to-device variation was found to have insignificant impact on the encryption process. Similarly, small variation in temperature is expected to have minimal impact on the proposed crypto engine.

The manuscript clearly states the context and gives explanations, however the adjective “biomimetic” in the title I personally would have omitted since it obscures the term “artificial neuron based on MoS₂ FETs” to form an AN encoder. So I would recommend the term “artificial neural encoder” instead but leave it up to the authors to decide.

We would like to thank the reviewer for appreciating our explanations. We agree with the reviewer that the term biomimetic is not appropriate, therefore we have replaced it with bioinspired. The revised title is “*A Low-Power Bio-inspired Crypto Engine for All-In-One IoT based on Programmable and Multifunctional MoS₂ FETs.*”

References, are modest, out of 21 ~6 are self-citations, so the list could be extended to include also other possibilities of IOT security, such as memory less systems e.g. with physical unclonable functions (PUFs) and optical ANs, also some more IoT security and neuromorphic computing

systems. e.g. M. Z. Alom and T. M. Taha, "Network intrusion detection for cyber security on neuromorphic computing system," 2017 International Joint Conference on Neural Networks (IJCNN), Anchorage, AK, 2017, pp. 3830-3837, doi: 10.1109/IJCNN.2017.7966339.

We are sorry for the missing references and would like to thank the reviewer for pointing them. We have tried our best to cite the recent and most relevant articles related to IoT security and neuromorphic related systems in the revised manuscript and supplementary information.

References

- [1] A. Oberoi, A. Dodda, H. Liu, M. Terrones, and S. Das, "Secure **Electronics** Enabled by Atomically Thin and Photosensitive Two-Dimensional Memtransistors," *ACS nano*, 2021.
- [2] W. Liu, Z. Wang, X. Liu, N. Zeng, Y. Liu, and F. E. Alsaadi, "A survey of deep neural network architectures and their applications," *Neurocomputing*, vol. 234, pp. 11-26, 2017.
- [3] U. Banerjee and A. P. Chandrakasan, "A Low-Power Elliptic Curve Pairing Crypto-Processor for Secure Embedded Blockchain and Functional Encryption," in *2021 IEEE Custom Integrated Circuits Conference (CICC)*, 2021: IEEE, pp. 1-2.
- [4] F. Alfaleh, H. Alfahaid, M. Alanzy, and S. Elkhediri, "Wireless sensor networks security: case study," in *2019 2nd International Conference on Computer Applications & Information Security (ICCAIS)*, 2019: IEEE, pp. 1-4.
- [5] M. Benaissa, "Throughput/area-efficient ECC processor using Montgomery point multiplication on FPGA," *IEEE Transactions on Circuits and Systems II: Express Briefs*, vol. 62, no. 11, pp. 1078-1082, 2015.
- [6] P. Choi, M.-K. Lee, J.-H. Kim, and D. K. Kim, "Low-complexity elliptic curve cryptography processor based on configurable partial modular reduction over NIST prime fields," *IEEE Transactions on Circuits and Systems II: Express Briefs*, vol. 65, no. 11, pp. 1703-1707, 2017.
- [7] Z. U. Khan and M. Benaissa, "High-Speed and Low-Latency ECC Processor Implementation Over GF (2^m) on FPGA," *IEEE Transactions on Very Large Scale Integration (VLSI) Systems*, vol. 25, no. 1, pp. 165-176, 2016.
- [8] J. Ding, S. Li, and Z. Gu, "High-speed ECC processor over NIST prime fields applied with Toom–Cook multiplication," *IEEE Transactions on Circuits and Systems I: Regular Papers*, vol. 66, no. 3, pp. 1003-1016, 2018.
- [9] J.-W. Lee, S.-C. Chung, H.-C. Chang, and C.-Y. Lee, "Efficient power-analysis-resistant dual-field elliptic curve cryptographic processor using heterogeneous dual-processing-element architecture," *IEEE Transactions on very large scale integration (vlsi) systems*, vol. 22, no. 1, pp. 49-61, 2013.
- [10] P. Kocher, J. Jaffe, and B. Jun, "Differential power analysis," in *Annual international cryptology conference*, 1999: Springer, pp. 388-397.

- [11] P. C. Kocher, "Timing attacks on implementations of Diffie-Hellman, RSA, DSS, and other systems," in *Annual International Cryptology Conference*, 1996: Springer, pp. 104-113.
- [12] O. Lopez-Sanchez, D. Lembke, M. Kayci, A. Radenovic, and A. Kis, "Ultrasensitive photodetectors based on monolayer MoS₂," *Nat Nanotechnol*, vol. 8, no. 7, pp. 497-501, Jul 2013, doi: 10.1038/nnano.2013.100.
- [13] R. Ge *et al.*, "Atomristor: Nonvolatile Resistance Switching in Atomic Sheets of Transition Metal Dichalcogenides," *Nano Lett*, vol. 18, no. 1, pp. 434-441, Jan 10 2018, doi: 10.1021/acs.nanolett.7b04342.
- [14] C. Liu, X. Yan, X. Song, S. Ding, D. W. Zhang, and P. Zhou, "A semi-floating gate memory based on van der Waals heterostructures for quasi-non-volatile applications," *Nat Nanotechnol*, vol. 13, no. 5, pp. 404-410, May 2018, doi: 10.1038/s41565-018-0102-6.
- [15] M. Wang *et al.*, "Robust memristors based on layered two-dimensional materials," *Nat Electron*, vol. 1, no. 2, pp. 130-136, 2018, doi: 10.1038/s41928-018-0021-4.
- [16] S. Bertolazzi, D. Krasnozhon, and A. Kis, "Nonvolatile memory cells based on MoS₂/graphene heterostructures," *ACS Nano*, vol. 7, no. 4, pp. 3246-52, Apr 23 2013, doi: 10.1021/nn3059136.
- [17] M. S. Choi *et al.*, "Controlled charge trapping by molybdenum disulphide and graphene in ultrathin heterostructured memory devices," *Nat Commun*, vol. 4, p. 1624, 2013, doi: 10.1038/ncomms2652.
- [18] Q. A. Vu *et al.*, "Two-terminal floating-gate memory with van der Waals heterostructures for ultrahigh on/off ratio," *Nat Commun*, vol. 7, p. 12725, Sep 2 2016, doi: 10.1038/ncomms12725.
- [19] K. Roy *et al.*, "Graphene-MoS₂ hybrid structures for multifunctional photoresponsive memory devices," *Nat Nanotechnol*, vol. 8, no. 11, pp. 826-30, Nov 2013, doi: 10.1038/nnano.2013.206.
- [20] S. Seo *et al.*, "Artificial optic-neural synapse for colored and color-mixed pattern recognition," *Nat Commun*, vol. 9, no. 1, p. 5106, Nov 30 2018, doi: 10.1038/s41467-018-07572-5.

- [21] S. Wang *et al.*, "A MoS₂ /PTCDA Hybrid Heterojunction Synapse with Efficient Photoelectric Dual Modulation and Versatility," *Adv Mater*, vol. 31, no. 3, p. e1806227, Jan 2019, doi: 10.1002/adma.201806227.
- [22] D. Lee *et al.*, "Multibit MoS₂ Photoelectronic Memory with Ultrahigh Sensitivity," *Adv Mater*, vol. 28, no. 41, pp. 9196-9202, Nov 2016, doi: 10.1002/adma.201603571.
- [23] D. Xiang *et al.*, "Two-dimensional multibit optoelectronic memory with broadband spectrum distinction," *Nat Commun*, vol. 9, no. 1, p. 2966, Jul 27 2018, doi: 10.1038/s41467-018-05397-w.
- [24] Q. Wang *et al.*, "Nonvolatile infrared memory in MoS₂/PbS van der Waals heterostructures," *Sci Adv*, vol. 4, no. 4, p. eaap7916, Apr 2018, doi: 10.1126/sciadv.aap7916.
- [25] H. Tian *et al.*, "A Dynamically Reconfigurable Ambipolar Black Phosphorus Memory Device," *ACS Nano*, vol. 10, no. 11, pp. 10428-10435, Nov 22 2016, doi: 10.1021/acsnano.6b06293.
- [26] D. Li, X. Wang, Q. Zhang, L. Zou, X. Xu, and Z. Zhang, "Nonvolatile Floating-Gate Memories Based on Stacked Black Phosphorus-Boron Nitride-MoS₂Heterostructures," *Advanced Functional Materials*, vol. 25, no. 47, pp. 7360-7365, 2015, doi: 10.1002/adfm.201503645.
- [27] Y. T. Lee *et al.*, "Nonvolatile Charge Injection Memory Based on Black Phosphorous 2D Nanosheets for Charge Trapping and Active Channel Layers," *Advanced Functional Materials*, vol. 26, no. 31, pp. 5701-5707, 2016, doi: 10.1002/adfm.201602113.
- [28] J. Wang *et al.*, "Floating gate memory-based monolayer MoS₂ transistor with metal nanocrystals embedded in the gate dielectrics," *Small*, vol. 11, no. 2, pp. 208-13, Jan 14 2015, doi: 10.1002/sml.201401872.
- [29] A. Dodda, A. Oberoi, A. Sebastian, T. H. Choudhury, J. M. Redwing, and S. Das, "Stochastic resonance in MoS₂ photodetector," *Nat Commun*, vol. 11, no. 1, p. 4406, Sep 2 2020, doi: 10.1038/s41467-020-18195-0.
- [30] D. Jayachandran *et al.*, "A low-power biomimetic collision detector based on an in-memory molybdenum disulfide photodetector," *Nat Electron*, vol. 3, no. 10, pp. 646-655, 2020, doi: 10.1038/s41928-020-00466-9.

- [31] A. Lipatov, P. Sharma, A. Gruverman, and A. Sinitskii, "Optoelectrical Molybdenum Disulfide (MoS₂)--Ferroelectric Memories," *ACS Nano*, vol. 9, no. 8, pp. 8089-98, Aug 25 2015, doi: 10.1021/acsnano.5b02078.
- [32] L. Mennel, J. Symonowicz, S. Wachter, D. K. Polyushkin, A. J. Molina-Mendoza, and T. Mueller, "Ultrafast machine vision with 2D material neural network image sensors," *Nature*, vol. 579, no. 7797, pp. 62-66, Mar 2020, doi: 10.1038/s41586-020-2038-x.
- [33] A. A. Bessonov, M. N. Kirikova, D. I. Petukhov, M. Allen, T. Ryhanen, and M. J. Bailey, "Layered memristive and memcapacitive switches for printable electronics," *Nat Mater*, vol. 14, no. 2, pp. 199-204, Feb 2015, doi: 10.1038/nmat4135.
- [34] B. Shao *et al.*, "Crypto primitive of MOCVD MoS₂ transistors for highly secured physical unclonable functions," *Nano Research*, 2020, doi: 10.1007/s12274-020-3033-0.
- [35] Z. Wang *et al.*, "Memristors with diffusive dynamics as synaptic emulators for neuromorphic computing," *Nat Mater*, vol. 16, no. 1, pp. 101-108, Jan 2017, doi: 10.1038/nmat4756.
- [36] P. Yao *et al.*, "Face classification using electronic synapses," *Nat Commun*, vol. 8, p. 15199, May 12 2017, doi: 10.1038/ncomms15199.
- [37] S. Wu *et al.*, "Bipolar resistance switching in transparent ITO/LaAlO(3)/SrTiO(3) memristors," *ACS Appl Mater Interfaces*, vol. 6, no. 11, pp. 8575-9, Jun 11 2014, doi: 10.1021/am501387w.
- [38] T. Ohno, T. Hasegawa, T. Tsuruoka, K. Terabe, J. K. Gimzewski, and M. Aono, "Short-term plasticity and long-term potentiation mimicked in single inorganic synapses," *Nat Mater*, vol. 10, no. 8, pp. 591-5, Jun 26 2011, doi: 10.1038/nmat3054.
- [39] O. Ionescu, C. Besleaga, V. Dumitru, and E. Pricop, "UAV identification system based on memristor physical unclonable functions," in *2020 12th International Conference on Electronics, Computers and Artificial Intelligence (ECAI)*, 2020: IEEE, pp. 1-4.
- [40] H. Nili *et al.*, "Hardware-intrinsic security primitives enabled by analogue state and nonlinear conductance variations in integrated memristors," *Nat Electron*, vol. 1, no. 3, pp. 197-202, 2018, doi: 10.1038/s41928-018-0039-7.
- [41] R. Zhang *et al.*, "Nanoscale diffusive memristor crossbars as physical unclonable functions," *Nanoscale*, vol. 10, no. 6, pp. 2721-2726, Feb 8 2018, doi: 10.1039/c7nr06561b.

- [42] H. Jiang *et al.*, "A novel true random number generator based on a stochastic diffusive memristor," *Nat Commun*, vol. 8, no. 1, p. 882, Oct 12 2017, doi: 10.1038/s41467-017-00869-x.
- [43] N. Ge *et al.*, "An efficient analog Hamming distance comparator realized with a unipolar memristor array: a showcase of physical computing," *Sci Rep*, vol. 7, p. 40135, Jan 5 2017, doi: 10.1038/srep40135.
- [44] J. Feldmann, N. Youngblood, C. D. Wright, H. Bhaskaran, and W. H. Pernice, "All-optical spiking neurosynaptic networks with self-learning capabilities," *Nature*, vol. 569, no. 7755, pp. 208-214, 2019.
- [45] L. Yin *et al.*, "Synaptic silicon-nanocrystal phototransistors for neuromorphic computing," *Nano Energy*, vol. 63, p. 103859, 2019.

REVIEWER COMMENTS

Reviewer #1 (Remarks to the Author):

The authors have addressed most of the questions in the previous review report except the following questions:

1. During sensing the current, the VN6 is kept at 0V. Will the LED light change the conductance of the TWGNA and disturb the pre-programmed random noise conductance value?
2. In such CBA, the VN6 nodes in the same column are connected, indicating that the TWGNA in the same column should be programmed simultaneously. In other words, it is impossible to program each TWGNA individually in such an array. In this case, the authors need to explain how they define the standard deviation and evaluate how much such column-by-column programming can influence the randomness of the VPSV.
3. To substantiate the claim on the advantages offered by their platform, it will be more convincing if the authors can prove that their platform is capable of delivering better performance metrics (in terms of energy consumption, footprint, memory endurance, encryption security, etc) as compared to other state-of-the-art platforms.

Reviewer #3 (Remarks to the Author):

The authors have done a good job in addressing the reviewers comments. However, some open questions remain. Specifically, I would like to comment on the responses to reviewer 3 and add a few more points concerning the revised manuscript.

comment 1:

The authors have now included a clear explanation and circuit level description in text and Figures R13 and R14. Also the monolithic system is now much clearer to the reader. Please note that in the Figure caption of Fig. R14 e) is not labelled in the image.

I believe the fundamental work on transistor level is very close to earlier works by the authors (ACS Nano 15, 16172–16182 (2021), ACS Nano 15, 19815–19827 (2021) and ACS Nano 15, 17804–17812 (2021)). However, I see that the circuit architecture with the 5T cells is new and not discussed in the prior works. My impression is that the hardware and circuit design have been substantially changed during the process of revising the manuscript. This gives me the impression that there were some shortcomings, in particular in the memory periphery. Only now the architecture allows for sensing, memory and encryption. The sensing is restricted to optical stimuli in the current version of the platform which may impose a limitation concerning the scope of applications.

comment 2: It is true that there was a misunderstanding. The generated white noise from the TWGNA is emulating the noisy synapsis. My point was more on the robustness of this architecture with respect to various noise sources.

comment 3: The benchmark table shows, that mostly the achievement of this paper is the monolithic integration of all three functions (sensing, storage and encryption). While this is true in general, I feel that each single domain may struggle to be competitive, especially when looking at the level of encryption. As also remarked by reviewer 2, the level of security is far below the standard 128 Bit encryption for state-of-the-art AES encryption. The authors should comment on the min entropy of their system and the scalability of their platform in that respect.

Comment 4: Concerning the low power argument it is hard to compare any figure of merits to a benchmark systems from the manuscript text. For example one could compare the power needed to encrypt and decrypt a certain information (bit width). I believe authors should define some metric to back up their argument of low power and what they compare to.

Reviewer #4 (Remarks to the Author):

I was asked by the editor to specifically evaluate the security claims in this manuscript.

First, I thank the authors for addressing the comments by a reviewer regarding the security properties of the proposed sensor. The reference presented are a good summary of recent work on public-key cryptographic implementations. It may be useful to point out that there is a standardization underway in the area of lightweight symmetric-key cryptography, driven by the National Institute of Standards and Technology (NIST). However, this effort in LWC emphasizes algorithmic, rather than technological solutions. In that sense, the proposed sensor technique in this manuscript is still very relevant and potentially groundbreaking.

The paper can still benefit from a careful definition of the security properties of the proposed sensor. While the authors now use the term 'information encryption', that term seems still too general to capture what is going on. The information security community has defined terms to describe the security properties of information, including 'Confidentiality' (the information is secure from eavesdroppers), 'Integrity' (the information is secure from tampering), 'Message Authenticity' (the origin of the information can be guaranteed). It seems that the proposed solution aims at 'Confidentiality' but not any of the other properties.

I am not sure that the proposed image encoding can be properly called an 'encryption' mechanism, because it appears no secret knowledge is needed to 'decrypt' the information. The decryption step consists of combining multiple 'encrypted' messages and taking a majority vote over the multiple of received images. Any receiver, also an eavesdropping receiver, appears to be capable of this step. The authors have assumed that multiple independent channels will be used to transmit multiple encoded versions of the same image to the true receiver. An eavesdropping receiver would only be at a disadvantage if that receiver is unable to check the same set of channels as the true receiver. But that is a significant assumption, that is underlying the security level of the proposed sensor. The common term to describe the security of such a noisy communication system is called a 'wiretap channel'.

The authors present a security demonstration based on decoding a noisy image with a neural network, but in my understanding each of these decodings is based on a single image at a time. There exists no decoder, not even one that has perfect knowledge of the encoder, that could perfectly restore the image based on a single observation, because some of the pixels are buried in noise for any single image instance. In that sense, the demonstration with the neural network appears to generate a predictable outcome (noise in results in noise out).

The above observations do not imply that the proposed circuit has no contributions to secure IoT; but it does imply that the security claims made by the authors have to be stated carefully. I'd like to verify with the authors that they intend to build a true encryption system or a wiretap channel. I would disagree with the notion of an encryption system, but agree with the notion of a wiretap channel.

It may also be helpful to define the 'attacker model' more clearly. For example, the attacker model used in the neural network attack is a different one from the model you use to demonstrate image decoding. In each case there are a different number of channels, different number of images at play.

Reviewer #1 (Remarks to the Author):

The authors have addressed most of the questions in the previous review report except the following questions

We are glad that the reviewer is satisfied with our response. We are happy to address the remaining concerns.

1. During sensing the current, the VN6 is kept at 0V. Will the LED light change the conductance of the TWGNA and disturb the pre-programmed random noise conductance value?

Reviewer's concern is noted. We have exploited photogating effect in the MoS₂ FET to sense the optical information. Therefore, when $V_{\text{illum}} > 0$ V, i.e. illuminations in the on-state and the subthreshold regime, the ratio of post-illumination conductance to dark conductance (r_{PH}), $r_{\text{PH}} = 1$, whereas, for $V_{\text{illum}} < 0$ V, i.e. illuminations in the off-state, $r_{\text{PH}} \gg 1$. Since the node voltage, $V_{\text{N6}} = 0$ V, is in the subthreshold regime of T_{WGNA} , we observe no change in the conductance T_{WGNA} when exposed to light. We have made this clarification in the revised manuscript.

2. In such CBA, the VN6 nodes in the same column are connected, indicating that the TWGNA in the same column should be programmed simultaneously. In other words, it is impossible to program each TWGNA individually in such an array. In this case, the authors need to explain how they define the standard deviation and evaluate how much such column-by-column programming can influence the randomness of the VPSV.

Reviewer's concern is noted. It is true that the V_{N6} nodes in a column are all connected, however note that each device in the column have separate source and drain contacts. So, unless the source and drain terminals are grounded it is difficult to program any device in a given column even though they share the same gate terminal. This enables us to program each device individually and achieve different standard deviation values (σ_G).

3. To substantiate the claim on the advantages offered by their platform, it will be more convincing if the authors can prove that their platform is capable of delivering better performance metrics (in

terms of energy consumption, footprint, memory endurance, encryption security, etc) as compared to other state-of-the-art platforms.

Reviewer's concern is noted. We have performed extensive literature survey and included the following discussion in the revised manuscript. Kindly note that a detailed quantitative comparison is difficult owing to the diverse nature of the applications and experimental setups, however, a qualitative assessment can be presented.

There has been several reports on lightweight cryptographic primitives that offer adequate security measures for resource-constrained devices. This includes block and stream ciphers as well as hash functions. For example, Bahrami *et. al* proposed a lightweight stream algorithm for the encryption of gray-scale images.¹ While the proposed scheme can be implemented on a 32-bit microcontroller,² hardware demonstration is yet to be realized. pure software implementation and is not suitable for hardware applications. Symmetric encryption is the most popular design choice for image encryption. The advanced encryption standard (AES) is a classic example of symmetric encryption. AES is an energy-efficient encryptor and a widely employed security system.³ However the AES algorithm is computationally expensive and is hard to implement at the sensor node to manage real-time applications. Some of the other lightweight schemes include CLEFIA⁴, PRESENT⁵, PRINCE⁶, and SIMON and SPECK⁷, which can be implemented with a very small area and power using serial data paths. However, these serial architectures lag in performance and do not essentially provide energy optimal operation. Moreover, serial architectures are highly susceptible to side-channel attacks.⁸ In addition to the attacks, the amount of energy consumed in the above-mentioned lightweight cryptographic primitives ranges from ~ 10 nanojoules to ~ 100 microjoules.⁹ Recently, Pham *et. al* have proposed an architecture to construct visual sensor motes employing FPGA (field-programmable gate array) platform. This architecture is predicted to be 20 times more efficient than other common visual sensor motes but consumes ~2 mJ of energy.¹⁰ Similarly, Aziz *et. al* proposed a hardware architecture for efficient DWT (discrete wavelet transform) coding of sensed images selectively using the JPEG2000 codec. This architecture consumes ~34 mW of power in which ~3 mW of power is consumed in DWT for image processing and compression. Sensor motes with 16-bit processor may perform efficient sensing and transmission functions, but specialized modules for image processing are required for higher efficiency, which leads to energy and area overhead.¹¹⁻¹³ The key contribution of our work is

developing a crypto engine that integrates sensing, storage, compute, and security at miniscule energy expenditure of ~ 10 nanoJoules. We would also like to mention that the majority of the hardware-implemented sensor node cryptography do not mention any information regarding the size of the transistor as they use FPGA and conventional silicon chips;, so it is difficult to comment on the size and endurance metrics.

Reviewer #3 (Remarks to the Author):

The authors have done a good job in addressing the reviewers comments. However, some open questions remain. Specifically, I would like to comment on the responses to reviewer 3 and add a few more points concerning the revised manuscript.

We would like to thank the reviewer for appreciating our efforts in addressing reviewer's comments. We have addressed the remaining concerns that the reviewer has raised.

Comment 1: The authors have now included a clear explanation and circuit level description in text and Figures R13 and R14. Also the monolithic system is now much clearer to the reader. Please note that in the Figure caption of Fig. R14 e) is not labelled in the image.

We are sorry for the typographical error in the rebuttal.

I believe the fundamental work on transistor level is very close to earlier works by the authors (ACS Nano 15, 16172–16182 (2021), ACS Nano 15, 19815–19827 (2021) and ACS Nano 15, 17804–17812 (2021)). However, I see that the circuit architecture with the 5T cells is new and not discussed in the prior works. My impression is that the hardware and circuit design have been substantially changed during the process of revising the manuscript. This gives me the impression that there were some shortcomings, in particular in the memory periphery. Only now the architecture allows for sensing, memory and encryption. The sensing is restricted to optical stimuli in the current version of the platform which may impose a limitation concerning the scope of applications.

Reviewer's observation is correct. Fundamental work on transistor level has some similarity to our earlier works published in ACS Nano 15, 19815–19827 (2021) and ACS Nano 15, 17804–17812 (2021). However, the work presented on Camouflaging in ACS Nano 15, 16172–16182 (2021) has no overlap with the current work. The reviewer is also correct that for monolithic integration of sensing, memory and encryption, and multi-pixel demonstration, we had to redesign the circuit.

Note that, one of the main motivation behind using two-dimensional (2D) layered MoS₂ as a hardware platform for our crypto engine is its multifunctional capability as sensor. Beyond photodetectors¹⁴, MoS₂ based FETs have been used as chemical sensors¹⁵, biological sensors¹⁵, touch sensors¹⁶, and radiation sensors.¹⁷ Therefore, the sensing unit of our MoS₂-based crypto engine is not limited to only optical stimuli. We have highlighted this aspect in the revised manuscript.

Comment 2: It is true that there was a misunderstanding. The generated white noise from the TWGNA is emulating the noisy synapses. My point was more on the robustness of this architecture with respect to various noise sources.

We apologize for the misunderstanding. One of the major advantages of our proposed T_{WGNA} is the tunable mean and standard deviation.¹⁸ Since we can program and erase our T_{WGNA} to multiple conductance states, we can achieve precise control over the mean and standard deviations of noise. However, in the conventional noise sources, which rely on mobility or carrier fluctuations in the device originating from the device dimensions, doping, contacts, etc.^{19,20} it is extremely difficult to have precise control over these parameters. This makes our proposed architecture more robust to unwanted noise sources compared to conventional designs.

Comment 3: The benchmark table shows, that mostly the achievement of this paper is the monolithic integration of all three functions (sensing, storage and encryption). While this is true in general, I feel that each single domain may struggle to be competitive, especially when looking at the level of encryption. As also remarked by reviewer 2, the level of security is far below the standard 128 Bit encryption for state-of-the-art AES encryption. The authors should comment on the min entropy of their system and the scalability of their platform in that respect.

Reviewer has raised a valid concern regarding the number of bits. In our revised full-scale demonstration, the maximum detectivity obtained from the 8×8 cryptographic engine was observed to be ~ 0.29 at a standard deviation (σ_G) of 0.4. This corresponds to a BFTs of $\sim 3.3 \times 10^{34}$ or 115 bits of security. The BFTs from the crypto engine can be further enhanced by encoding at

different standard deviations. Furthermore, the number of BFTs increases exponentially with increasing bit length (S) as shown in Fig. R1.

Entropy is another critical parameter determining the resilience of the cryptographic platform. The entropy of the crypto engine is determined by the programming capability of the T_{WGNA} . Wali *et al* obtained close to ideal entropy¹⁸ by exploiting the programming capability of the MoS₂ FETs.

Finally our platform is aggressively scalable thanks to the atomically thin body nature of monolayer MoS₂. Recent studies show high performance monolayer MoS₂ FETs with the channel and contact lengths scaled to 29 nm and 13 nm, respectively.²¹ Moreover, some of the early criticism of 2D FETs have also been successfully addressed in recent years through the realization of low contact resistance,²² high ON current,²³ integration of ultra-thin and high-k gate dielectric,²⁴ and wafer scale growth using chemical vapor deposition (CVD) and metal organic CVD (MOCVD).^{25,26} Similarly, MoS₂ based microprocessors,²⁷ analogue operational amplifier,²⁸ RF electronics components,²⁹ and neuromorphic and biomimetic hardware platforms,³⁰⁻³² have been reported.

Figure R1. Number of brute force trials (BFTs) required to decode the letter 'N' for different image size (S) as a function of σ_G .

Comment 4: Concerning the low power argument it is hard to compare any figure of merits to a benchmark systems from the manuscript text. For example one could compare the power needed to encrypt and decrypt a certain information (bit width). I believe authors should define some metric to back up their argument of low power and what they compare to.

Reviewer's concern is noted. We have performed extensive literature survey and included the following discussion in the revised manuscript. Kindly note that a detailed quantitative comparison is difficult owing to the diverse nature of the applications and experimental setups, however, a qualitative assessment can be presented.

There has been several reports on lightweight cryptographic primitives that offer adequate security measures for resource-constrained devices. This includes block and stream ciphers as well as hash functions. For example, Bahrami *et. al* proposed a lightweight stream algorithm for the encryption of gray-scale images.¹ While the proposed scheme can be implemented on a 32-bit microcontroller,² hardware demonstration is yet to be realized. pure software implementation and is not suitable for hardware applications. Symmetric encryption is the most popular design choice for image encryption. The advanced encryption standard (AES) is a classic example of symmetric encryption. AES is an energy-efficient encryptor and a widely employed security system.³ However the AES algorithm is computationally expensive and is hard to implement at the sensor node to manage real-time applications. Some of the other lightweight schemes include CLEFIA⁴, PRESENT⁵, PRINCE⁶, and SIMON and SPECK⁷, which can be implemented with a very small area and power using serial data paths. However, these serial architectures lag in performance and do not essentially provide energy optimal operation. Moreover, serial architectures are highly susceptible to side-channel attacks.⁸ In addition to the attacks, the amount of energy consumed in the above-mentioned lightweight cryptographic primitives ranges from ~ 10 nanojoules to ~ 100 microjoules.⁹ Recently, Pham *et. al* have proposed an architecture to construct visual sensor motes employing FPGA (field-programmable gate array) platform. This architecture is predicted to be 20 times more efficient than other common visual sensor motes but consumes ~ 2 mJ of energy.¹⁰ Similarly, Aziz *et. al* proposed a hardware architecture for efficient DWT (discrete wavelet transform) coding of sensed images selectively using the JPEG2000 codec. This architecture consumes ~ 34 mW of power in which ~ 3 mW of power is consumed in DWT for image processing and compression. Sensor motes with 16-bit processor may perform efficient sensing and transmission functions, but specialized modules for image processing are required for higher efficiency, which leads to energy and area overhead.¹¹⁻¹³ The key contribution of our work is developing a crypto engine that integrates sensing, storage, compute, and security at miniscule energy expenditure of ~ 10 nanoJoules. We would also like to mention that the majority of the hardware-implemented sensor node cryptography do not mention any information regarding the size of the transistor as they use FPGA and conventional silicon chips, so it is difficult to comment on the size and endurance metrics.

Reviewer #4 (Remarks to the Author):

I was asked by the editor to specifically evaluate the security claims in this manuscript.

First, I thank the authors for addressing the comments by a reviewer regarding the security properties of the proposed sensor. The reference presented are a good summary of recent work on public-key cryptographic implementations. It may be useful to point out that there is a standardization underway in the area of lightweight symmetric-key cryptography, driven by the National Institute of Standards and Technology (NIST). However, this effort in LWC emphasizes algorithmic, rather than technological solutions. In that sense, the proposed sensor technique in this manuscript is still very relevant and potentially groundbreaking.

We would like to thank the reviewer for considering our work to be relevant and potentially groundbreaking. Following reviewer's suggestion, we have now included a statement mentioning the standardization of light weight cryptography (LWC) driven by National Institute of Standards and Technology (NIST) in the revised manuscript.

The paper can still benefit from a careful definition of the security properties of the proposed sensor. While the authors now use the term 'information encryption', that term seems still too general to capture what is going on. The information security community has defined terms to describe the security properties of information, including 'Confidentiality' (the information is secure from eavesdroppers), 'Integrity' (the information is secure from tampering), 'Message Authenticity' (the origin of the information can be guaranteed). It seems that the proposed solution aims at 'Confidentiality' but not any of the other properties.

We agree with the reviewer that our proposed security scheme is inclined towards the confidentiality of the end user. We have mentioned the same in the revised manuscript.

I am not sure that the proposed image encoding can be properly called an 'encryption' mechanism, because it appears no secret knowledge is needed to 'decrypt' the information. The decryption step consists of combining multiple 'encrypted' messages and taking a majority vote over the multiple of received images. Any receiver, also an eavesdropping receiver, appears to be capable of this

step. The authors have assumed that multiple independent channels will be used to transmit multiple encoded versions of the same image to the true receiver. An eavesdropping receiver would only be at a disadvantage if that receiver is unable to check the same set of channels as the true receiver. But that is a significant assumption, that is underlying the security level of the proposed sensor. The common term to describe the security of such a noisy communication system is called a 'wiretap channel'.

Reviewer's observation is correct. The proposed security model is similar to the wiretap channel.³³ We have mentioned the underlying assumption in the revised manuscript.

The authors present a security demonstration based on decoding a noisy image with a neural network, but in my understanding each of these decodings is based on a single image at a time. There exists no decoder, not even one that has perfect knowledge of the encoder, that could perfectly restore the image based on a single observation, because some of the pixels are buried in noise for any single image instance. In that sense, the demonstration with the neural network appears to generate a predictable outcome (noise in results in noise out).

We agree with the reviewer that the encoded images are difficult to decode using single observation. However, the idea behind neural network was to emphasize on the fact that even a trained neural network would lead to a lower inference accuracy for the encoded images based on our proposed encoding scheme. We have mentioned explicitly the attack model in our revised manuscript.

The above observations do not imply that the proposed circuit has no contributions to secure IoT; but it does imply that the security claims made by the authors have to be stated carefully. I'd like to verify with the authors that they intend to build a true encryption system or a wiretap channel. I would disagree with the notion of an encryption system but agree with the notion of a wiretap channel.

We would like to thank the reviewer for his comments and valuable inputs. We have restated our security claims abiding to the strict mathematical definition in security field and we agree with the reviewer that our proposed security scheme is wiretap channel.

It may also be helpful to define the 'attacker model' more clearly. For example, the attacker model used in the neural network attack is a different one from the model you use to demonstrate image decoding. In each case there are a different number of channels, different number of images at play.

Reviewer's concern is noted. The attack model exploited in our proposed cryptographic scheme is "wiretapping". However, the idea behind neural network was to emphasize on the fact that even a trained neural network would lead to a lower inference accuracy for the encoded images based on the proposed encoding scheme. We have mentioned explicitly the attack model in our revised manuscript.

References

- 1 Bahrami, S. & Naderi, M. Image encryption using a lightweight stream encryption algorithm. *Advances in Multimedia* **2012** (2012).
- 2 Janakiraman, S., Thenmozhi, K., Rayappan, J. B. B. & Amirtharajan, R. Lightweight chaotic image encryption algorithm for real-time embedded system: Implementation and analysis on 32-bit microcontroller. *Microprocessors and Microsystems* **56**, 1-12 (2018).
- 3 Gonçalves, D. D. O. & Costa, D. G. A survey of image security in wireless sensor networks. *Journal of Imaging* **1**, 4-30 (2015).
- 4 Proença, P. & Chaves, R. in *2011 21st International Conference on Field Programmable Logic and Applications*. 512-517 (IEEE).
- 5 Bogdanov, A. *et al.* in *International workshop on cryptographic hardware and embedded systems*. 450-466 (Springer).
- 6 Borghoff, J. *et al.* in *International conference on the theory and application of cryptology and information security*. 208-225 (Springer).
- 7 Beaulieu, R. *et al.* in *Proceedings of the 52nd Annual Design Automation Conference*. 1-6.
- 8 Bogdanov, A. in *International Workshop on Selected Areas in Cryptography*. 84-95 (Springer).
- 9 Alioto, M. Trends in hardware security: From basics to ASICs. *IEEE Solid-State Circuits Magazine* **11**, 56-74 (2019).
- 10 Pham, D. M. & Aziz, S. M. Object extraction scheme and protocol for energy efficient image communication over wireless sensor networks. *Computer Networks* **57**, 2949-2960 (2013).
- 11 Pham, D. M. & Aziz, S. M. in *2013 IEEE Eighth International Conference on Intelligent Sensors, Sensor Networks and Information Processing*. 260-264 (IEEE).
- 12 Hasan, K. K., Ngah, U. K. & Salleh, M. F. M. Efficient hardware-based image compression schemes for wireless sensor networks: A survey. *Wireless personal communications* **77**, 1415-1436 (2014).
- 13 Pham, D. M. & Aziz, S. M. in *2011 IFIP 9th International Conference on Embedded and Ubiquitous Computing*. 100-105 (IEEE).
- 14 Yin, Z. *et al.* Single-layer MoS₂ phototransistors. *ACS nano* **6**, 74-80 (2012).
- 15 Wang, L. *et al.* Functionalized MoS₂ nanosheet-based field-effect biosensor for label-free sensitive detection of cancer marker proteins in solution. *Small* **10**, 1101-1105 (2014).
- 16 Park, M. *et al.* MoS₂-based tactile sensor for electronic skin applications. *Advanced Materials* **28**, 2556-2562 (2016).
- 17 Arnold, A. J., Shi, T., Jovanovic, I. & Das, S. Extraordinary Radiation Hardness of Atomically Thin MoS₂. *ACS applied materials & interfaces* **11**, 8391-8399 (2019).
- 18 Wali, A., Ravichandran, H. & Das, S. A Machine Learning Attack Resilient True Random Number Generator Based on Stochastic Programming of Atomically Thin Transistors. *ACS Nano* **15**, 17804-17812, doi:10.1021/acsnano.1c05984 (2021).
- 19 Renteria, J. *et al.* Low-frequency 1/f noise in MoS₂ transistors: Relative contributions of the channel and contacts. *Applied Physics Letters* **104**, 153104 (2014).
- 20 Kwon, J., Delker, C. J., Thomas Harris, C., Das, S. R. & Janes, D. B. Experimental and modeling study of 1/f noise in multilayer MoS₂ and MoSe₂ field-effect transistors. *Journal of Applied Physics* **128**, 094501 (2020).

- 21 Smets, Q. *et al.* in *2019 IEEE International Electron Devices Meeting (IEDM)*. 23.22. 21-23.22. 24 (IEEE).
- 22 Rai, A. *et al.* Air Stable Doping and Intrinsic Mobility Enhancement in Monolayer Molybdenum Disulfide by Amorphous Titanium Suboxide Encapsulation. *Nano Lett* **15**, 4329-4336, doi:10.1021/acs.nanolett.5b00314 (2015).
- 23 English, C. D., Smithe, K. K. H., Xu, R. L. & Pop, E. in *2016 IEEE International Electron Devices Meeting (IEDM)*. 5.6.1-5.6.4.
- 24 Price, K. M., Schauble, K. E., McGuire, F. A., Farmer, D. B. & Franklin, A. D. Uniform Growth of Sub-5-Nanometer High- κ Dielectrics on MoS₂ Using Plasma-Enhanced Atomic Layer Deposition. *ACS applied materials & interfaces* **9**, 23072-23080 (2017).
- 25 2DCC. *2d-crystal-consortium*, <<https://www.mri.psu.edu/2d-crystal-consortium/user-facilities/thin-films/list-thin-film-samples-available>> (
- 26 Kang, K. *et al.* High-mobility three-atom-thick semiconducting films with wafer-scale homogeneity. *Nature* **520**, 656 (2015).
- 27 Wachter, S., Polyushkin, D. K., Bethge, O. & Mueller, T. A microprocessor based on a two-dimensional semiconductor. *Nature communications* **8**, 14948 (2017).
- 28 Polyushkin, D. K. *et al.* Analogue two-dimensional semiconductor electronics. *Nature Electronics* **3**, 486-491, doi:10.1038/s41928-020-0460-6 (2020).
- 29 Gao, Q. *et al.* Scalable high performance radio frequency electronics based on large domain bilayer MoS₂. *Nature Communications* **9**, 4778, doi:10.1038/s41467-018-07135-8 (2018).
- 30 Das, S., Dodda, A. & Das, S. A biomimetic 2D transistor for audiomorphic computing. *Nature Communications* **10**, 3450, doi:10.1038/s41467-019-11381-9 (2019).
- 31 Sebastian, A., Pannone, A., Radhakrishnan, S. S. & Das, S. Gaussian synapses for probabilistic neural networks. *Nature communications* **10**, 1-11 (2019).
- 32 Arnold, A. J. *et al.* Mimicking Neurotransmitter Release in Chemical Synapses via Hysteresis Engineering in MoS₂ Transistors. *ACS nano* **11**, 3110-3118 (2017).
- 33 Wyner, A. D. The wire-tap channel. *Bell system technical journal* **54**, 1355-1387 (1975).

REVIEWERS' COMMENTS

Reviewer #1 (Remarks to the Author):

The authors have addressed the remaining questions satisfactorily. I recommend its publication in Nature Communications.

Reviewer #3 (Remarks to the Author):

The authors have addressed all of my concerns satisfactorily.

Reviewer #4 (Remarks to the Author):

Thanks for your revision.

The revisions are acceptable, and thanks for confirming that your security model corresponds to a wiretap channel - a knowledgeable security researcher will now be able to judge the encryption method to its actual capabilities.